# Generative Models are Self-Watermarked: Intellectual Property Declaration through Re-Generation

## Abstract

Protecting intellectual property for generated data has emerged as a critical concern for AI corporations, as machine-generated content proliferates. Reusing generated data without permission poses a formidable barrier to safeguarding the intellectual property tied to these models. The verification of data ownership is further complicated by the use of Machine Learning as a Service (MLaaS), which often operates as a black-box system.

Our work is dedicated to detecting data reuse from even an individual sample. In contrast to watermarking techniques that embed additional information as watermark triggers into models or generated content, our approach does not introduce artificial watermarks which may compromise the quality of model outputs. Our investigation reveals the existence of latent fingerprints inherently present within deep learning models. In response, we propose an explainable verification procedure to verify data ownership through re-generation. Furthermore, we introduce a novel methodology to amplify the model fingerprints through iterative data re-generation and a theoretical grounding on the proposed approach. We demonstrate the viability of our approach using recent advanced text and image generative models.

[Notes for the revision]: **inserted/responding text** and ~~deleted text~~.

## 1 Introduction

In recent years, Artificial Intelligence Generated Content (AIGC) has gained widespread recognition for the quality of its generated content. Many companies have opted to offer their well-trained generative models as pay-as-you-use services through their cloud platforms, including examples like ChatGPT, Claude, DALL-E, Stable Diffusion, Copilot, *etc.* However, the accessibility of these models has not only accelerated the dissemination and progress of AI technology but has also given rise to concerns regarding model misuse and piracy. Protecting the intellectual property (IP) of these generated contents poses significant challenges, as users can readily distribute these contents for their own interest. This includes actions such as selling the data to other parties without authorization, entering art contests with the generated contents to vie for prizes, and even extracting and publishing surrogate models to compete as service rivals.

The primary technical approach for safeguarding IP is to embed subtle but verifiable watermarks into the generated content, such as text (He et al., 2022a;b; Kirchenbauer et al., 2023), images (Zear et al., 2018; Zhao et al., 2023b) and code (Lee et al., 2023). However, traditional watermarking techniques typically introduce supplementary information, either implicitly to the deep learning model's parameters and architectures or directly to the generated outputs through post-processing. It is worth noting that any alternations made to the generative model or its outputs could potentially result in a decline in the quality of the generated contents. The alternative approach involves classifying data generated by a model to distinguish it from content produced by other models or human (Solaiman et al., 2019; Ippolito et al., 2020). Nonetheless, many of these solutions require training additional models, specifically classifiers, to verify authorship. Another potential concern is the ability of these additional classifiers to generalize and maintain robustness across emerging generative models with limited training resources.

**I. Generation Stage**     **II. Verification Stage**

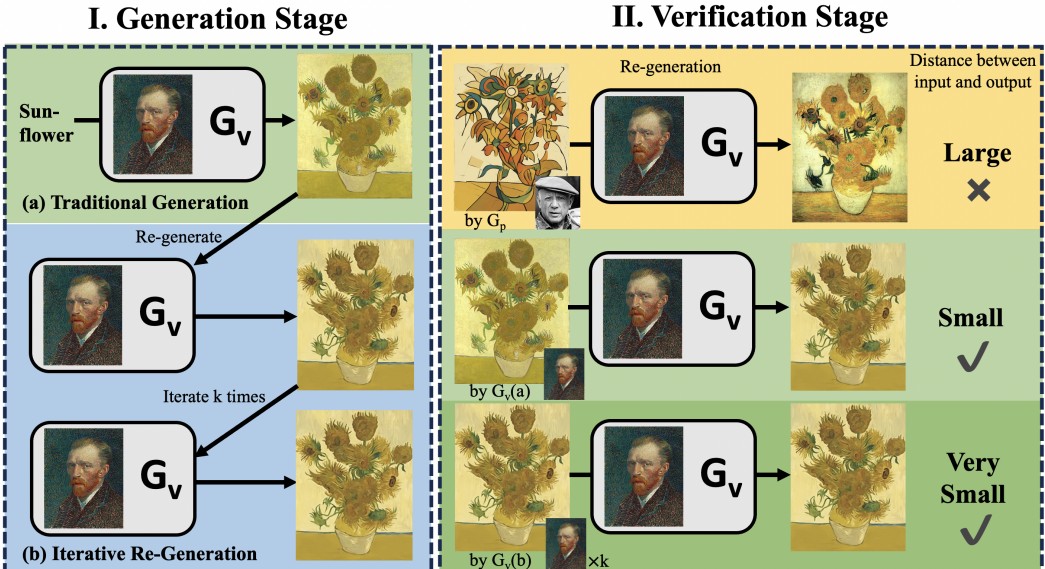

Figure 1: The framework of using intrinsic fingerprints in generative models. In (I) *Generation Stage*, models generate outputs in traditional ways or iteratively re-generate the intended outputs before publication. In (II) *Verification Stage*, data ownership is certified by comparing the distance of the data with the re-generated data. $\mathcal{G}_v$ and $\mathcal{G}_p$ represent models with the styles of Van Gogh ($v$) and Pablo Picasso ($p$) respectively.

In light of the aforementioned challenges, we initially note that generative models inherently possess their unique attributes, such as styles and knowledge, akin to model fingerprints. These implicit fingerprints can be verified by comparing genuine data samples with the re-generated content. Secondly, these implicit fingerprints in the generative models can be effectively reinforced by employing the authentic model to iteratively re-generate the intended outputs from prior iterations, leveraging the fixed-point theory (Granas & Dugundji, 2003). In Figure 1, we present a schematic representation of the framework for generating and verifying model fingerprints. Consider, for instance, an author such as Van Gogh ($\mathcal{G}_v$) who wishes to establish the authorship of his artwork. In this context, it would be inherently easier for Van Gogh to produce a painting bearing close resemblance to his original than for other artists, for example, Pablo Picasso ($\mathcal{G}_p$), to emulate the same piece. Furthermore, the fingerprints, which include attributes like painting style and composition, can be iteratively enhanced by refining or revising the artwork.

We summarize the key advantages and contributions of our work as follows:

- We demonstrate the effectiveness of re-generated data as the crucial signals for verifying the authorship, applicable in black-box settings for both Computer Vision (CV) and Natural Language Processing (NLP) applications.

- We introduce an iterative re-generation method to enhance the fingerprints within generative models. The fixed-point theory is used to prove that the modification through re-generation will converge to small edit distances, allowing for clear differentiation from data generated by other models.

- We propose a practical verification protocol to validate data ownership in generative models, eliminating the need for generators to disclose their model parameters or watermarking rules during legal proceedings.

- Our approach relies solely on the standard generative models without the need for (1) manipulating or fine-tuning generative model parameters, (2) applying post-processing to the model outputs, or (3) introducing additional independent (classification) models for verification purposes.

## 2 RELATED WORK

**IP protection for image generation models**   Recent advancements in the field of generative modeling, exemplified by innovations like DALL·E 2 (Ramesh et al.) and Stable Diffusion (Rombach et al., 2022), have facilitated the creation and manipulation of photo-realistic images. Concurrently, this surge in synthetic images has also given rise to ethical concerns regarding potential misuse, including issues such as deep fakes, impersonation, and copyright infringement (Brundage et al., 2018; Rostamzadeh et al., 2021). Consequently, there has been a growing emphasis on tracing and verifying the origins of such images. One established method for ascertaining image ownership and ensuring copyright protection is watermarking. Traditional watermarking techniques involve direct modifications to pixel values, *e.g.,* in spatial domain, or embedding watermarks within transformed versions of the image, *e.g.,* in frequency domain (Cox et al., 2008).

With the advancements of deep learning techniques, neural networks have been utilized for image generation (Goodfellow et al., 2014; Kingma & Welling, 2014). Building on this, Zhu et al. (2018) have suggested leveraging neural networks to seamlessly encode concealed information within images in a fully trainable manner. Inspired by this idea, Fernandez et al. (2022) incorporate watermarks into the latent spaces formulated by a self-supervised network like DINO (Caron et al., 2021). This approach modulates the features of the image within a specific region of the latent space, ensuring that subsequent transformations applied to watermarked images preserve the integrity of the embedded information. Subsequently, watermark detection can be conducted within this same latent space. Similarly, Fernandez et al. (2023) introduces a binary signature directly into the decoder of a diffusion model, resulting in images containing an imperceptibly embedded binary signature. This binary signature can be accurately extracted using a pre-trained watermark extractor during the verification process.

**IP protection for text generation models**   Likewise, content generated by text generation models is increasingly vulnerable to various forms of misuse, including the spread of misinformation (Hazell, 2023; Mozes et al., 2023; Sjouwerman, 2023) and the training of surrogate models (Wallace et al., 2020; Xu et al., 2022). Consequently, there has been a growing interest in protecting the IP of text generation models through the use of watermarks. However, unlike images, textual information is composed of discrete tokens, making the watermarking process for text a difficult endeavor due to the potential for inadvertent alternation that can change its semantic meaning (Katzenbeisser & Petitolas, 2000). One solution for preserving semantic integrity during watermarking involves synonym substitution (Topkara et al., 2006; Chang & Clark, 2014; He et al., 2022a). Nevertheless, the simplistic approach to synonym substitution is vulnerable to detection through statistical analyses. In response, He et al. (2022b) have proposed a conditional synonym substitution method to enhance both the stealthiness and robustness of substitution-based watermarks. Moreover, Venugopal et al. (2011) adopted bit representation to encode semantically similar sentences, enabling the selection of watermarked sentences through bit manipulation. With the advent of neural language models, traditional rule-based watermarking has evolved towards neural methodologies. For instance, Kirchenbauer et al. (2023) successfully watermark output sentences by prompting the model to yield more watermarked tokens. Rather than producing discrete watermarked tokens, one can modify the output distribution to embed a subtle watermark within a continuous space (Zhao et al., 2023a).

## 3 METHODOLOGY

In our research, our primary focus is on the threat posed by malicious users who engage in unauthorized usage of generated content. Specifically, we examine the scenario where the owner of authentic generative models, referred to as $\mathcal{G}_a$, grants access to their models in a black-box fashion, permitting users to query their API for content generation. Unfortunately, there exists a risk that these malicious users may exploit the generated content $x$ without acknowledging the generator's license. Furthermore, they might even falsely attribute the content to unauthorized parties, represented as $\mathcal{G}_\times$. To address this issue, API providers can actively monitor the characteristics of publicly available data to identify potential cases of plagiarism. **This can be accomplished by applying the re-generation and measuring the corresponding distance in editing, as described in Section 3.1. To enhance verification accuracy, $\mathcal{G}_a$ can employ an iterative re-generation approach to bolster the finger-**

**printing signal, as introduced and proved in Sections 3.2. If there are suspicions of plagiarism, the company can initiate legal proceedings against the alleged plagiarist through a third-party arbitration, following the protocol proposed in Section 3.3.**

### 3.1 AUTHORSHIP VERIFICATION THROUGH RE-GENERATION

The authentic generative model $\mathcal{G}_a$ aims to distinguish between the data samples it generated, denoted as $x_a$, with the benign samples $x_\times$ generated by other models $\mathcal{G}_\times$ for contrast. To verify the data, the authentic model *i)* re-generates the data $\mathcal{G}_a(\boldsymbol{x})$ and *ii)* evaluates the distance between the original sample and the re-generated sample, defined as $d(\boldsymbol{x}, \mathcal{G}) \triangleq \mathbb{D}(\mathcal{G}(\boldsymbol{x}), \boldsymbol{x})$. In essence, during the re-generation process, samples produced by the authentic model are expected to exhibit lower 'self-edit' distance, as they share the same generative model $\mathcal{G}_a$, which uses identical knowledge, such as writing or painting styles, effectively serving as model fingerprints. In mathematical terms, we have

$$\mathbb{D}(\mathcal{G}_a(\boldsymbol{x}_a), \boldsymbol{x}_a) < \mathbb{D}(\mathcal{G}_a(\boldsymbol{x}_\times), \boldsymbol{x}_\times), \ \ i.e., \ d(\boldsymbol{x}_a, \mathcal{G}_a) < d(\boldsymbol{x}_\times, \mathcal{G}_a). \tag{1}$$

Consequently, authentic models can identify their own samples by evaluating this 'self-edit' distance, which can be viewed as a specialized form of a classification function for discriminating the authentic and contrast models. Additionally, the re-generation process and corresponding 'edit' distances can serve as explainable and comprehensible evidence to human judges.

### 3.2 ENHANCING FINGERPRINT THROUGH ITERATIVE RE-GENERATION

While we have claimed that the data samples generated through the vanilla generative process can be verified, there is no theoretical guarantee regarding the 'self-edit' distance for certifying these samples using the authentic model. To address this limitation, we introduce an iterative re-generation method that improves the fingerprinting capability of the authentic generative model, **as it manages to reduce the 'self-edit' distances. This property will be utilised in verification.**. This section provides a theoretical foundation for examining the characteristics of re-generated samples by exploring the convergence behavior of the fixed points by iterative functions.

**Definition 1 (The Fixed Points of a Lipschitz Continuous Function)** *Given a multi-variable function $f : \mathbb{R}^m \rightarrow \mathbb{R}^m$ is $L$-Lipschitz continuous, i.e., $\|f(\boldsymbol{x}) - f(\boldsymbol{y})\| \leq L \cdot \|\boldsymbol{x} - \boldsymbol{y}\|$ where $L \in (0, 1)$. For any initial point $\boldsymbol{x}_0$, the sequence $\{\boldsymbol{x}_i\}_{i=0}^\infty$ is acquired by the recursion $\boldsymbol{x}_{i+1} = f(\boldsymbol{x}_i)$. The sequence converges to a fixed point $\boldsymbol{x}_*$, where $\boldsymbol{x}_* = f(\boldsymbol{x}_*)$.*

**Theorem 1 (The Convergence of Step Distance for $k$-th Re-generation)** *The distance between the input and output of the $k$-th iteration is bounded by*

$$\|\boldsymbol{x}_{k+1} - \boldsymbol{x}_k\| \leq L^k \cdot \|\boldsymbol{x}_1 - \boldsymbol{x}_0\|, \tag{2}$$

*and the distance converges to 0 given $L \in (0, 1)$.*

**Proof** *We apply $L$-Lipschitz continuous property recursively,*

$$\begin{aligned} \|\boldsymbol{x}_{k+1} - \boldsymbol{x}_k\| &= \|f(\boldsymbol{x}_k) - f(\boldsymbol{x}_k - 1)\| \\ &\leq L \cdot \|\boldsymbol{x}_k - \boldsymbol{x}_{k-1}\| = L \cdot \|f(\boldsymbol{x}_{k-1}) - f(\boldsymbol{x}_k - 2)\| \\ &\leq L^2 \cdot \|\boldsymbol{x}_{k-1} - \boldsymbol{x}_{k-2}\| \leq \cdots \leq L^k \cdot \|\boldsymbol{x}_1 - \boldsymbol{x}_0\|. \end{aligned} \tag{3}$$

**Theorem 2 (Banach Fixed-Point Theorem)** *(Banach, 1922; Ciesielski, 2007) The theory can be extended to distance metrics in Banach space. Let $(\mathcal{X}, \mathbb{D})$ be a complete metric space, and $f : \mathcal{X} \rightarrow \mathcal{X}$ is a contraction mapping, if there exists $L \in (0, 1)$ such that for all $\boldsymbol{x}, \boldsymbol{y} \in \mathcal{X}$*

$$\mathbb{D}(f(\boldsymbol{x}), f(\boldsymbol{y})) \leq L \cdot \mathbb{D}(\boldsymbol{x}, \boldsymbol{y}). \tag{4}$$

*Then, we have the following convergence of the distance sequence similar to Theorem 1,*

$$\mathbb{D}(\boldsymbol{x}_{k+1} - \boldsymbol{x}_k) \leq L^k \cdot \mathbb{D}(\boldsymbol{x}_1 - \boldsymbol{x}_0). \tag{5}$$

### 3.3 DATA GENERATION AND VERIFICATION PROTOCOL

Hereby, the defense protocol is comprised of two key components: *i)* Iterative Generation Algorithm 1, which progressively enhances the fingerprint signal in the generated outputs; and *ii)* Verification Algorithm 2 is responsible for confirming the authorship of the data sample through a

one-step re-generation process using both authentic model $\mathcal{G}_a$ and suspected contrast model $\mathcal{G}_\times$, with a confidence margin $\delta$.

| **Algorithm 1** Data Generation Algorithm. | **Algorithm 2** Verification Algorithm. |
|---|---|
| **Input:** Prompt $\boldsymbol{p}$ as the requirement for generation and number of iteration $k$.
**Output:** Image, Text, etc. with enhanced fingerprints in the generative model.
1: $\boldsymbol{x} \leftarrow \mathcal{G}(\boldsymbol{p})$ ▷ Initial generation.
2: **for** $i \leftarrow 1$ to $k$ **do** ▷ Iterate $k$ steps.
3:  $\boldsymbol{x} \leftarrow \mathcal{G}(\boldsymbol{x})$ ▷ Re-generation.
4: **end for**
5: **return** $\boldsymbol{x}$ | **Input:** Data sample $\boldsymbol{x}_a$ generated by $\mathcal{G}_a$ and misused by $\mathcal{G}_\times$.
**Output:** IP infringement by $\mathcal{G}_\times$.
1: $\boldsymbol{y}_a \leftarrow \mathcal{G}_a(\boldsymbol{x}_a)$ ▷ Regenerate data by model $\mathcal{G}_a$.
2: $\boldsymbol{y}_\times \leftarrow \mathcal{G}_\times(\boldsymbol{x}_a)$ ▷ Regenerate data by model $\mathcal{G}_\times$.
3: $r \leftarrow \mathbb{D}(\boldsymbol{y}_\times, \boldsymbol{x}_a)/\mathbb{D}(\boldsymbol{y}_a, \boldsymbol{x}_a)$
4: **return** $r > 1 + \delta$ |

**Imagine artists refining their distinct writing or painting styles during each artwork replication. Similarly, a generative model's unique 'style' becomes more defined during image re-generation, as deviations reduce. This mirrors iterative functions which converge to fixed points. In our case, each re-generation brings the image closer to the model's inherent style, and the distinct fingerprint facilitates AI authorship verification.**

## 4 EXPERIMENTS

This section aims to demonstrate the efficacy of our re-generation method on generative models for text and image separately. For both application scenarios, we first generate data samples with various number of re-generation steps using Algorithm 1. Then, we test the properties of these samples by three series of experiments. *i)* We verify the convergence of the distance between one-step re-generation (*Distance Convergence*). *ii)* We illustrate the discrepancy between the distances by the authentic models and the 'suspected' contrast models (*Discrepancy*). *iii)* We report the accuracy of correctly identified samples from the authentic model and the percentage of misclassified samples by contrast model based on Algorithm 2 (*Verification*).

### 4.1 EXPERIMENTAL SETUP

**Generative Models.** For text generation, we consider four generative models: 1) **M2M** (Fan et al., 2021): a multilingual encoder-decoder model trained for many-to-many multilingual translation; 2) **mBART-large-50** (Tang et al., 2021): a model fine-tuned on mBART for multilingual machine translation between any pair of 50 languages; 3) **GPT3.5-turbo**: a chat-based GPT3.5 model developed by OpenAI[1]; 4) **Cohere**: a large language model developed by Cohere.

For image generation, we examine five primary generative models based on the Stable Diffusion (SD) architecture (Rombach et al., 2022). All models are trained on a subset of the LAION-2B dataset (Schuhmann et al., 2022) consisting of CLIP-filtered image-text pairs. These models are: 1) **SDv2.1**; 2) **SDXLv1.0**, which distinguishes itself by employing an ensemble of experts pipeline for latent diffusion, where the base model first generates noisy latent that are refined using a specialized denoising model (Podell et al., 2023); 3) **SDv2**; 4) **SDv2.1 Base**; 5) **SDXL Base0.9** (Podell et al., 2023).[2] More information on model architecture and training, and the quality of re-generated images is provided in Appendix B.1.

**Distance Metrics.** To gauge the similarity between inputs and outputs, we employ three popular similarity metrics for NLP experiments, 1) **BLEU** (Papineni et al., 2002); 2) **ROUGE-L** (Lin, 2004) and 3) **BERTScore** (Zhang et al., 2020). We transform all similarity scores $s \in [0, 1]$ to distances $d$ by $d = 1 - s$. For CV experiments, we consider the following distance metrics: 1) **CLIP Cosine Distance** measures the semantic similarity using pre-trained CLIP image-text embeddings, where a lower distance indicates more closely aligned high-level content (Radford et al., 2021); 2) **LPIPS**

---

[1]We have also studied GPT4 and present its corresponding results in the Appendix D.2.

[2]Additionally, we explored other variants, including **SDv1.5**, **SDv1.4**, and **Nota-AI's BK-SDM Base**. The corresponding experiments and results are provided in Appendix B.1.

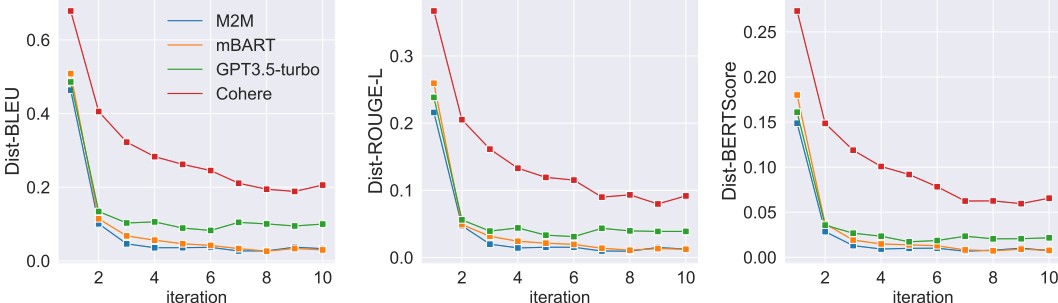

Figure 2: The convergence analysis of the distances in iteration based on various metrics on the re-generated text of 200 samples from in-house datasets.

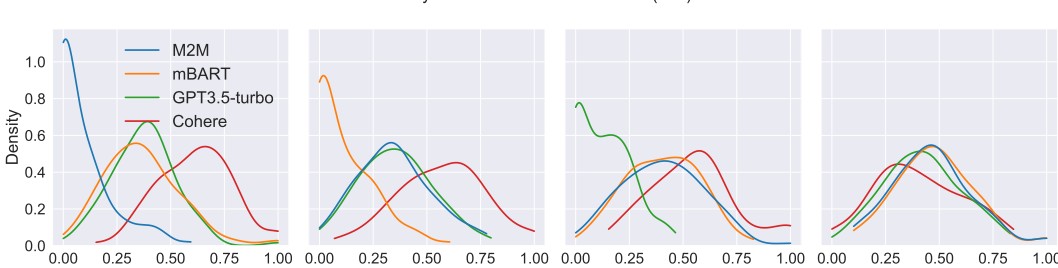

Figure 3: Density distribution of one-step re-generation using four text generation models. The authentic models from left to right are 1) M2M, 2) mBART, 3) GPT3.5-turbo, and 4) Cohere.

compares perceptual style differences (Zhang et al., 2018); 3) **Mean Squared Error (MSE)** serves as a pixel-level metric that compares raw image values; 4) **Structural Similarity Index (SSIM)** assesses image degradation based on luminance, contrast, and structure (Wang et al., 2004).

Note that we gauge the success of our approach based on the satisfactory performance of any of these metrics when applied to the verification of generative models.

## 4.2 NATURAL LANGUAGE GENERATION EXPERIMENTS

In the study of text generation, the primary objective is to translate a French sentence into English. Then, we leverage round-trip translation to paraphrase the English sentences, which serves as *re-generation*. Specifically, for each round-trip translation step, an English input sentence is first translated into French then it is translated back to English. We repeat this process multiple times to observe the change of the 'edit' distances. For GPT3.5-turbo and Cohere, we perform all experiments in a zero-shot setting. It has been known that text generative models exhibit superior performance when test data inadvertently overlaps with pre-training data, a phenomenon referred to as data contamination (Magar & Schwartz, 2022). To address this potential issue, we sample 200 sentences from the in-house data as the starting point for each model. Comprehensive settings for this generation process are provided in Appendix C.

**Distance Convergence.** The dynamics of the distance change across three metrics and various generative models are depicted in Figure 2. We observe remarkable reduction in distances between the first and second iterations in all settings. Subsequently, the changes in distance between consecutive iterations exhibit a diminishing pattern, tending to converge after approximately 5-7 rounds of re-generation. These observed trends are consistent with the fixed-point theorem as elaborated in Section 3.2.

**Discrepancy.** In our study of iterative re-generation using the same model, we observe convergence which can be utilized to distinguish the authentic model from its counterparts. As delineated in Section 3.1, for each sentence $x$ from our corpus $X$, we apply the authentic model to yield $x_a$

Table 1: The accuracy of differentiating contrast models from the authentic model (M2M) using different $\delta$. ROUGE and BERT mean ROUGE-L and BERTScore, respectively.

| $\delta$ | mBART | | | GPT3.5-turbo | | | Cohere | | |
|---|---|---|---|---|---|---|---|---|---|
| | BLEU | ROUGE | BERT | BLEU | ROUGE | BERT | BLEU | ROUGE | BERT |
| 0.05 | 94.0 | 92.0 | 91.0 | 93.0 | 89.0 | 92.0 | 99.0 | 99.0 | 99.0 |
| 0.10 | 91.0 | 92.0 | 90.0 | 92.0 | 88.0 | 92.0 | 98.0 | 99.0 | 99.0 |
| 0.20 | 91.0 | 92.0 | 90.0 | 91.0 | 87.0 | 90.0 | 98.0 | 99.0 | 99.0 |
| 0.40 | 88.0 | 88.0 | 85.0 | 87.0 | 85.0 | 85.0 | 95.0 | 97.0 | 99.0 |

Table 2: The accuracy (Acc.) and misclassification rate (Mis.) of verifying the authentic models ($\mathcal{G}_a$) using different contrast models ($\mathcal{G}_\times$).

| $\mathcal{G}_a$ \ $\mathcal{G}_\times$ | M2M | | mBART | | GPT3.5-turbo | | Cohere | |
|---|---|---|---|---|---|---|---|---|
| | Acc. ↑ | Mis. ↓ | Acc. ↑ | Mis. ↓ | Acc. ↑ | Mis. ↓ | Acc. ↑ | Mis. ↓ |
| (a) $k = 1$ | | | | | | | | |
| M2M | - | - | 94.0 | 2.0 | 93.0 | 5.0 | 99.0 | 1.0 |
| mBART | 85.0 | 11.0 | - | - | 89.0 | 8.0 | 100.0 | 0.0 |
| GPT3.5-turbo | 87.0 | 8.0 | 90.0 | 8.0 | - | - | 99.0 | 1.0 |
| Cohere | 60.0 | 36.0 | 63.0 | 35.0 | 51.0 | 43.0 | - | - |
| (b) $k = 3$ | | | | | | | | |
| M2M | - | - | 94.0 | 1.0 | 95.0 | 3.0 | 100.0 | 0.0 |
| mBART | 85.0 | 7.0 | - | - | 90.0 | 4.0 | 100.0 | 0.0 |
| GPT3.5-turbo | 89.0 | 8.0 | 93.0 | 5.0 | - | - | 97.0 | 2.0 |
| Cohere | 75.0 | 17.0 | 73.0 | 24.0 | 63.0 | 31.0 | - | - |
| (c) $k = 5$ | | | | | | | | |
| M2M | - | - | 94.0 | 2.0 | 95.0 | 1.0 | 98.0 | 1.0 |
| mBART | 91.0 | 4.0 | - | - | 88.0 | 6.0 | 100.0 | 0.0 |
| GPT3.5-turbo | 90.0 | 6.0 | 94.0 | 2.0 | - | - | 95.0 | 3.0 |
| Cohere | 71.0 | 19.0 | 83.0 | 13.0 | 69.0 | 27.0 | - | - |

via a translation. Both authentic and contrast models then perform a one-step re-generation of $x_a$ to obtain $y$. Finally, we can measure the distance between $x_a$ and $y$ as described in Section 3.1. According to Figure 3, the density distribution associated with the authentic model demonstrates a noticeable divergence from those of models for contrast, thus affirming the hypothesis discussed in Section 3.1. This consistent pattern holds when using ROUGE-L and BERTScore for distance measures, as evidenced by their respective density distributions presented in Appendix D.1.

As depicted in Figure 2, the distances observed in subsequent iterations of the same model experience a substantial reduction. This implies that using sentences generated in later iterations can enhance the model's fingerprint. In particular, we can derive $x_a$ from $k > 1$ iterations of $\mathcal{G}(\cdot)$. As shown in Appendix D.1, a larger value of $k$ (*e.g.,* 3 or 5) further accentuates the distinctiveness of the authentic model's density distribution in comparison to the contrasting models.

**Verification.** We employ Algorithm 2 to ascertain if a given sentence originates from the authentic models. The determination hinges on the threshold parameter, $\delta$. Thus, we designate M2M as the authentic model, while using mBART, GPT3.5-turbo, and Cohere as contrast models to determine the optimal value of $\delta$. As indicated in Table 1, a threshold of 0.05 certifies that the three contrast models can validate that more than 93% of the samples are derived from M2M. As anticipated, an increase in the value of $\delta$ augments the stringency criteria, resulting in a reduction in verification accuracy. Therefore, we fix $\delta$ at 0.05 for ensuing evaluations unless stated otherwise.

Until this point, our discussion has verified the samples $x$ produced by the authentic models. We are also interested in the proportion of samples generated by the contrast model but misclassified as originating from the authentic model. This proportion is referred to as the **misclassification rate**.

As indicated in Table 2, our methodology verifies the authorship with an accuracy above 85% and a misclassification rate of less than 10% on M2M, mBART, and GPT3.5-turbo. Moreover, both

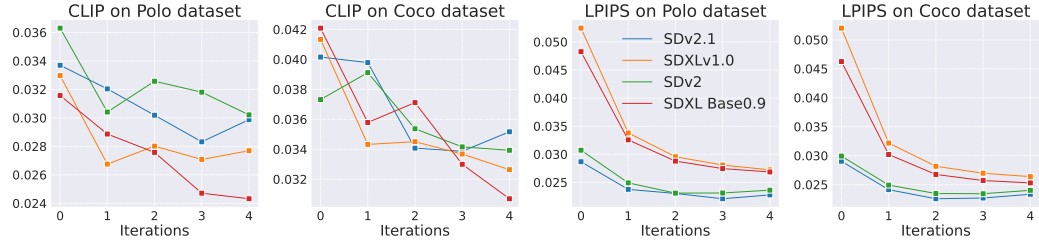

Figure 4: The convergence analysis of the distances in iteration based on various metrics on re-generated images of 200 samples on Coco and Polo datasets.

accuracy and misclassification metrics improve with additional re-generation iterations. While the authorship of Cohere can be ascertained, its performance lags behind the other models. We attribute the distinctive behavior of Cohere to its relatively slower convergence than other models.[3] Despite its slower convergence, its efficacy shows promise, particularly when the value of $k$ exceeds 3.

## 4.3 COMPUTER VISION EXPERIMENTS

The main objective of the image generation task is to produce an image given a text prompt using a generative model. To generate the initial proposal of images, we sample 200 prompts each from the **MS-COCO dataset (COCO)** (Lin et al., 2014) and **Polo Club Diffusion DB dataset (POLO)** (Wang et al., 2022), then we generate initial images corresponding to the prompts using all assigned models. We consider two settings in re-generating: *i)* watermarking images through in-painting and *ii)* fingerprinting via full image re-generation. The subsequent paragraphs detail the methodologies for each.

**Watermarking through In-painting** For a given image $x$, we mask $1/10$th of its pixels with fixed positions as the watermark pattern. The masked regions are then reconstructed using a generative model. Similar to text generation, the in-painting step is iterated multiple times, as depicted in Algorithm 1. A comprehensive description of the procedure is provided in Appendix A.1.

**Fingerprinting through Re-generation** In the re-generation setting, we first split all possible mask positions into 8 non-overlapped sets, coined *segments*, each with $1/8$ pixel positions for a given image $x_a$. We parallelly reconstruct each segment based on the rest of the image as context. Then, we reassemble the generated contents of all segments into a new image $y$. When analyzing reconstructions of $\mathcal{G}_a$ by contrasting models $\mathcal{G}_\times$, there is an expectation that models may exhibit variations in painting masked regions due to their inherent biases and training. By comparing $y$ to the original $x_a$, we aim to identify subtle model fingerprints based on the re-generated content. For this study, each model generates 200 images, which are then re-generated over five iterations. We compute LPIPS and CLIP similarity between $x_a$ and $y$ to quantify the model behavior.[4] The detailed discussion on the quality of re-generated images is provided in Appendix D.3.2.

**Distance Convergence.** Similarly to text re-generation, our method works for both watermarking and fingerprinting settings, as illustrated in Figures 8 and 4, indicated by consistent downward trends across datasets and distance metrics. This implies that the re-generation process converges, with subsequent iterations yielding images that more closely resemble their predecessors. While our methodology proves efficacious in both scenarios, subsequent sections will prioritize the fingerprinting setting. Details pertaining to the watermarking setting can be found in Appendix A.1.1.

**Discrepancy.** In our study, we focus on the setting of iterative re-generation through in-painting, we observe convergence that can reveal differences between models. Like the NLP section both authentic and contrast models perform a one-step re-generation of $x_a$ to obtain $y$ and the CLIP distance is measured between the outputs.

---

[3]We investigate the underlying reason behind the slow convergence for Cohere in Appendix D.3.1.

[4]The performance of MSE and SSIM is reported in Appendix A.2.1

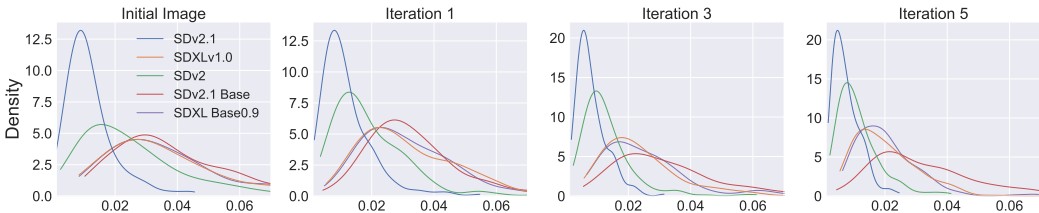

Figure 5: Verifying data generated by authentic $\mathcal{G}_a$ on Coco Dataset at various iterations.

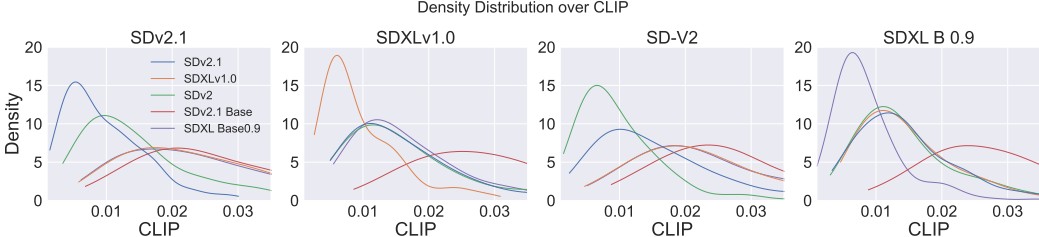

Figure 6: Density distribution of one-step re-generation among four image generation models on Polo Dataset. The authentic models from left to right are: 1) SD 2.1, 2) SDXL 1.0, 3) SD 2, 4) SDXL Base 0.9.

As illustrated in Figure 6, the one-step re-generation density distribution of the authentic model predominantly peaks at lower values across $\mathcal{G}$. This indicates that authentic models can effectively distinguish their own regenerated outputs from those of non-authentic models. The divergence in CLIP distances after multiple rounds of inpainting highlights our method's ability to distinguish between generative models. Using images from later in-painting iterations (*e.g.,* $k > 1$) further emphasizes each model's unique characteristics. A similar trend is evident with the Coco dataset and LPIPS distance metric, the details of which can be found in Appendix B.4.

**Verification.** In accordance with the text generation framework, we utilize Algorithm 2 to evaluate verification performance, selecting optimal parameters $k = 5$ and $\delta = 0.05$ based on the results of the text generation experiment. As demonstrated in Table 3, models reliably identify images from the authentic model, with accuracy often exceeding 85% and misclassification rates under 15%.

## 5 CONCLUSION

In this work, we observe the intrinsic fingerprint in both language and vision generative models, which can be identified by re-generating the data samples. Furthermore, we propose to enhance the fingerprints by iterative re-generation and prove the convergence of one-step re-generation distance for more convincing verification. Our research paves the way toward the IP protection protocol for generative models without i) modifying the original generative model, ii) post-processing to the generated outputs, or iii) an additional model for identity classification.

Table 3: The accuracy (Acc.) and misclassification rate (Mis.) of verifying the authentic models $\mathcal{G}_a$ using different contrast models $\mathcal{G}_\times$ on Polo Dataset for $k = 5$ and $\delta$ at 0.05.

| $\mathcal{G}_\times$ 
 $\mathcal{G}_a$ | SD v2.1 | | SD v2 | | SD v2.1B | | SDXL 0.9 | | SDXL 1.0 | |
|---|---|---|---|---|---|---|---|---|---|---|
| | Acc. ↑ | Mis. ↓ | Acc. ↑ | Mis. ↓ | Acc. ↑ | Mis. ↓ | Acc. ↑ | Mis. ↓ | Acc. ↑ | Mis. ↓ |
| **SD v2.1** | - | - | 80.0 | 14.0 | 99.5 | 0.5 | 98.5 | 0.5 | 99.0 | 0.0 |
| **SD v2** | 77.5 | 18.5 | - | - | 99.0 | 0.0 | 96.5 | 3.0 | 95.5 | 2.5 |
| **SD v2.1B** | 81.5 | 16.5 | 82.5 | 13.5 | - | - | 88.5 | 8.0 | 89.5 | 9.0 |
| **SDXL 0.9** | 97.5 | 2.5 | 96.5 | 2.0 | 99.5 | 0.5 | - | - | 92.0 | 6.5 |
| **SDXL 1.0** | 95.5 | 4.0 | 92.5 | 4.5 | 100.0 | 0.0 | 94.5 | 4.0 | - | - |

## REPRODUCIBILITY STATEMENT

In this section, we elucidate the reproducibility aspects of our research, encompassing both theoretical and empirical studies.

- **Theoretical Study**: Our methodology is grounded in the fixed-point theorem. Comprehensive proofs and associated algorithms can be found in Section 3.
- **Empirical Study**: For a thorough understanding of image and text generation, as well as re-generation procedures, please refer to Appendix A for image generation and Appendix C for text generation, respectively.

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

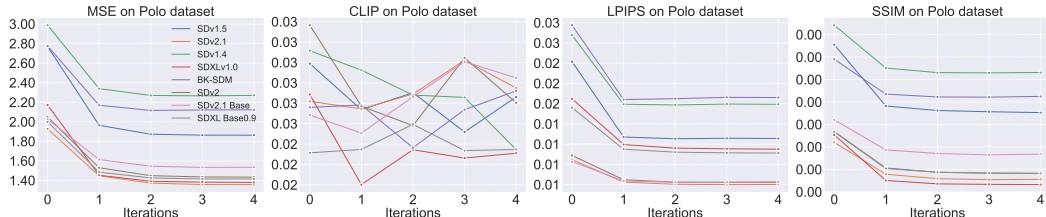

Figure 7: The convergence analysis of the distances in iteration based on various metrics on the watermarked images of 200 samples from Coco and Polo datasets.

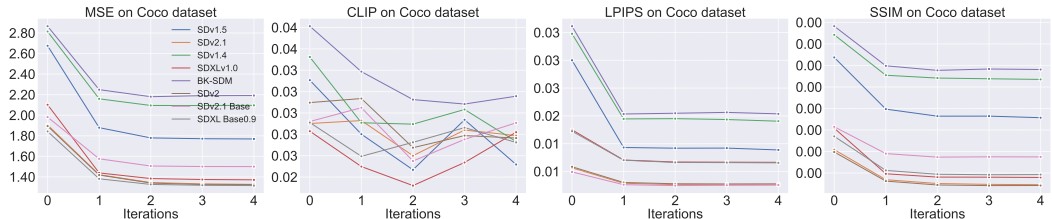

Figure 8: The convergence analysis of the distances in iteration based on various metrics on the watermarked images of 200 samples from Coco dataset

## A EXPERIMENTAL SETUP FOR IMAGE GENERATION

### A.1 EMBEDDING WATERMARK THROUGH INPAINTING

To analyze the stability and convergence properties of inpainting models, we perform an iterative masked image infilling procedure. Given an input image $x$ from model $\mathcal{G}_i$, we iteratively inpaint with mask $M$ using model $\mathcal{G}$:

$$x_{n+1} = \mathcal{G}(x_n, M)$$

Here, the mask $M$ not only guides the inpainting but also functions as the medium to embed our watermark. As we iteratively inpaint using a mask $M$, the watermark becomes more deeply embedded, serving as a distinctive signature to identify the source model $\mathcal{G}_i$.

### A.1.1 CONVERGENCE OF WATERMARKED IMAGES

The iterative masked inpainting procedure displays consistent convergence behavior across models. With a fixed binary mask covering $1/10$th of the image, the distance between successive image generations decreases rapidly over the first few iterations before stabilizing. This is evidenced by the declining trend in metrics like MSE, LPIPS, and CLIP similarity as iterations increase.

The early convergence suggests the generative models are effectively reconstructing the masked regions in a coherent manner. While perfect reconstruction is infeasible over many passes, the models appear to reach reasonable stability within 5 iterations as shown in Figure 7 and 8.

Convergence to a stable equilibrium highlights latent fingerprints in the model behavior. The consistent self-reconstruction statistics form the basis for distinguishing authentic sources in the subsequent fingerprinting experiments. The watermarking convergence analysis highlights model stability and confirms that iterative inpainting effectively removes embedded watermarks without degrading image quality (see Figures in D.3.2).

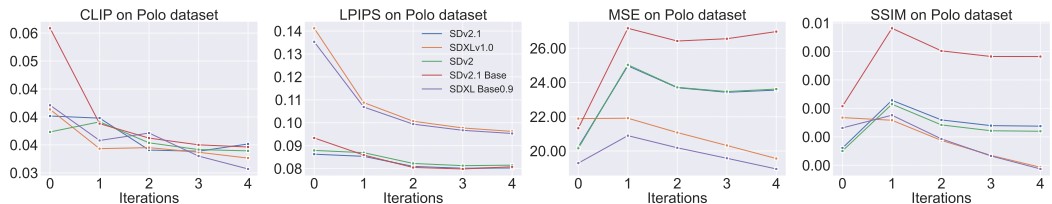

Figure 9: The convergence analysis of the distances in iteration based on various metrics on the regenerated images of 200 samples from Polo dataset.

Figure 10: The convergence analysis of the distances in iteration based on various metrics on the regenerated images of 200 samples from Coco dataset.

### A.2 ENHANCING AND VERIFYING FINGERPRINT THROUGH RE-GENERATION

We divide an image into non-overlapping segments and have models reconstruct masked regions to expose inherent fingerprints. Given an image $x$, we generate segment masks $M_1, ..., M_N$ covering $x$ and apply $\mathcal{G}_i$ to reconstruct each part:

$$y_k = \mathcal{G}_i(x, M_k)$$

By analyzing reconstructed segments $y_k$ and the composited image $y$, model-specific artifacts can be quantified without additional pattern information. A salient feature of our proposed regeneration paradigm is its independence from any additional information about the image partitioning pattern. The model's unique fingerprint emerges naturally during regeneration, regardless of how the image is divided or the components are merged. We experiments on 5 latest SD models - SDv 2.1, SD v2.1B, SD v2, SDXL 1.0, and SDXL0.9B using 8 segmented masks each covering $1/8$th of the image thereby totaling full image coverage. Model fingerprints are identified through LPIPS, and CLIP similarities between original $x_a$ and reconstructed $y$.

#### A.2.1 CONVERGENCE OF REGENERATED IMAGES

Enhancing fingerprint regeneration shows consistent convergence in perceptual metrics like CLIP and LPIPS within four iterations (see Figure 9 and 10). However, traditional metrics such as MSE and SSIM lack clear convergence, suggesting inpainting effectively captures visual content but not at the pixel level. The models converge to unique stable points, revealing inherent fingerprints based on their biases and training data. This divergence is important for model attribution. Overall, regeneration effectively exposes these fingerprints while maintaining visual integrity, underscoring perceptual superiority over pixel-based metrics in evaluating generative model fingerprints.

## B COMPUTER VISION SUPPLEMENTARY EXPERIMENTS AND DETAILS

### B.1 COMPUTER VISION GENERATIVE MODELS

We consider eight models based on the Stable Diffusion architecture (Rombach et al., 2022). These models leverage the architecture and differ primarily in their training schemes. The models are selected to span a range of architectures, training schemes, and dataset sizes. This diversity allows us to explore model-specific behaviors for attribution and stability analysis.

Table 4: Summary of models based on the Stable Diffusion architecture.

| Model | Description |
|-------|-------------|
| $Model_1$ | Stable Diffusion 1.5: Fine-tuned from SD v1.4 for 595k steps Rombach et al. (2022) |
| $Model_2$ | Stable Diffusion 2.1: Fine-tuned from SD v2 for 210k steps Rombach et al. (2022) |
| $Model_3$ | Stable Diffusion 1.4: Initialized from SD v1.4 weights |
| $Model_4$ | SDXL 1.0: Larger backbone, trained on LAION-5B Podell et al. (2023) |
| $Model_5$ | Block-removed Knowledge-distilled Stable Diffusion Model Kim et al. (2023) |
| $Model_6$ | Stable Diffusion 2 |
| $Model_7$ | Stable Diffusion v 2.1 Base |
| $Model_8$ | SDXL 0.9 Podell et al. (2023) |

All models support inpainting, allowing images to be edited given a mask and image. We utilize the inpainting pipeline - StableDiffussion and StableDiffusionXL - provided by HuggingFace (von Platen et al., 2022) [5](von Platen et al., 2022). Both SD v1.4 and SD v2 checkpoints were initialized with the weights of the SD v1.4 checkpoints and subsequently fine-tuned on "laion-aesthetics v2 5+". SDXL 1.0 employs a larger UNet backbone and a more potent text encoder (Podell et al., 2023). BKSDM is designed for efficient text-to-image synthesis, removing blocks from the U-Net and undergoing distillation pre-training on 0.22M LAION text-image pairs (Kim et al., 2023).

**Image Inpainting Image inpainting refers to filling in missing or masked regions of an image to reconstruct the original intact image. A fixed binary mask is applied to cover certain areas of the image. The binary masks are generated as blank images filled with random white pixels. To reconstruct images, the binary mask and original image are passed to the generative model's inpainting pipeline, where the values of the pixels in the masked areas are predicted based on the unmasked context. After inpainting, the reconstructed masked regions are merged back into the re-generated image for future iteration.**

### B.2 COMPUTER VISION DISTANCE METRICS

**Contrastive Language-Image Pretraining (CLIP) Cosine Distance**    The CLIP model encodes images into high-dimensional feature representations that capture semantic content and meaning (Radford et al., 2021). With pre-trained image-text models, we can easily capture the semantic similarity of two images, which is their shared meaning and content, regardless of their visual appearance.

CLIP uses a vision transformer model as the image encoder. The output of the final transformer block can be interpreted as a semantic feature vector describing the content of the image. Images with similar content will have feature vectors close together or aligned in the embedding space.

To compare two images I1 and I2 using CLIP, we can encode them into feature vectors $f1$ and $f2$. The cosine distance between these semantic feature vectors indicates the degree of semantic alignment.

$$\text{cosine\_sim}(f_1, f_2) = \frac{f_1 \cdot f_2}{\|f_1\|\|f_2\|} \tag{6}$$

$$\text{cosine\_dist}(f_1, f_2) = 1 - \text{cosine\_sim}(f_1, f_2) \tag{7}$$

Where $\cdot$ denotes the dot product and $\|\cdot\|$ denotes the $L_2$ norm. The cosine distance ranges from 0 to 1, with 0 indicating perfectly aligned features. A lower distance between images implies more similar high-level content and meaning in the images as captured by the CLIP feature embeddings. We specifically use OpenClip with a ConvNext-XXLarge encoder pretrained on laion2b dataset.

**Learned Perceptual Image Patch Similarity (LPIPS)**    LPIPS metric focuses on perceptual and stylistic similarities, by using a convolutional neural network pretrained on human judgements of

---

[5]**https://huggingface.co/docs/diffusers/api/pipelines/stable_diffusion/inpaint**

image patch similarities. The distance is measured between the CNN's intermediate feature representations of two images (Zhang et al., 2018).

$$d(x, x') = \sum_l \frac{1}{H_l W_l} \sum_{h,w} \left\| w_l \odot (\hat{y}_l^{hw} - \hat{y}_l'^{hw}) \right\|_2^2 \tag{8}$$

By comparing features across corresponding layers of the CNN, LPIPS can provide a fine-grained distance measuring subtle perceptual differences imperceptible in pixel space (Zhang et al., 2018). For example, changes to color scheme or artistic style that maintain semantic content will have a low CLIP distance but a higher LPIPS distance. We use the original implementation of Zhang (2023)

**Mean Squared Error (MSE)**   The mean squared error (MSE) between two images $x$ and $x'$ is calculated as:

$$MSE(x, x') = \frac{1}{mn} \sum_{i=0}^{m-1} \sum_{j=0}^{n-1} [x(i, j) - x'(i, j)]^2$$

Where $x$ and $x'$ are $m \times n$ images represented as matrices of pixel intensities. The MSE measures the average of the squared intensity differences between corresponding pixels in $x$ and $x'$.

It provides a simple pixel-level similarity metric sensitive to distortions like noise, blurring, and coloring errors. However, MSE lacks perceptual relevance and is not robust to geometric/structural changes in the image.

**Structural Similarity Index (SSIM)**   The structural similarity index (SSIM) (Hore & Ziou, 2010) compares corresponding $8 \times 8$ windows in the images across three terms - luminance, contrast, and structure:

$$SSIM(x, x') = \frac{(2\mu_x \mu_{x'} + c_1)(2\sigma_{xx'} + c_2)}{(\mu_x^2 + \mu_{x'}^2 + c_1)(\sigma_x^2 + \sigma_{x'}^2 + c_2)}$$

Where $\mu_x$, $\mu_{x'}$ are mean intensities, $\sigma_x^2$, $\sigma_{x'}^2$ are variances, and $\sigma_{xx'}$ is covariance for windows in $x$ and $x'$. $c_1$, $c_2$ stabilize division.

This decomposes similarity into comparative measurements of structure, luminance, and contrast. As a result, SSIM better matches human perceptual judgments compared to MSE. Values range from -1 to 1, with 1 being identical local structure.

As research rapidly improves generation quality, we expect future use cases to leverage such advanced generators. Analyzing these models is thus more indicative of real-world conditions going forward compared to earlier versions. Furthermore, the marked quality improvements in recent SD models present greater challenges for attribution and fingerprinting. Subtle inter-model differences become more difficult to quantify amidst high-fidelity outputs. Distance metrics like MSE and LPIPS are sensitive to quality, so lower baseline distortion is a more rigorous test scenario. By evaluating cutting-edge models without inpainting specialization, we aim to benchmark model fingerprinting efficacy on contemporary quality levels. Our experiments on the latest SD variants at scale also assess generalization across diverse high-fidelity generators. Successful attribution and stability analysis under these conditions will highlight the viability of our proposed techniques in real-world deployment.

### B.3   SAMPLE PROMPTS AND IMAGES REPRODUCED

In our computer vision experiments, we sample prompts from the MS-COCO 2017 Evaluation (Lin et al., 2014) and POLOCLUB (POLO) Diffusion Dataset (Wang et al., 2022) for image generation. We present a few example prompts and images produced for different models in the section below.

### B.3.1 COCO DATASET PROMPTS

- A motorcycle with its brake extended standing outside.

- Off white toilet with a faucet and controls.

- A group of scooters rides down a street.

### B.3.2 POLO DATASET PROMPTS

- A renaissance portrait of Dwayne Johnson, art in the style of Rembrandt!! Intricate. Ultra detailed, oil on canvas, wet-on-wet technique, pay attention to facial details, highly realistic, cinematic lightning, intricate textures, illusionistic detail.

- Epic 3D, become legend shiji! GPU mecha controlled by telepathic hackers, made of liquid, bubbles, crystals, and mangroves, Houdini SideFX, perfect render, ArtStation trending, by Jeremy Mann, Tsutomu Nihei and Ilya Kuvshinov.

- An airbrush painting of a cyber war machine scene in area 5 1 by Destiny Womack, Gregoire Boonzaier, Harrison Fisher, Richard Dadd.

### B.4 DENSITY DISTRIBUTION OF A ONE-STEP REGENERATION

The one-step regeneration density distributions reveal distinct model-specific characteristics, enabling discrimination as seen in Figure 11 and 12, Most models exhibit distinction, with each distribution showing unique traits.

A notable exception in this behavior is observed for the SD v2.1B (see Figures 11 and 12c). which initially demonstrates less discrimination. However, over extended iterations, SD v2.1B shows marked improvement, highlighting the capacity for iterative refinement. By the 5th iteration, there is a noticiable improvement in the discriminative nature of its one-step regeneration. This improvement is crucial as it highlights the model's capacity to refine and enhance its regenerative characteristics over time.

While LPIPS is not immediately effective in pinpointing the authentic model at the very first step, it still offers a powerful mechanism to distinguish between models. LPIPS is effective at differentiating between families of models, such as the Stable Diffusion models and the Stable Diffusion XL models, as visualized in Figure 13 for the Polo dataset and Figure 14 for the Coco dataset. The lack of effectiveness of LPIPS in identifying the authentic model is a primary reason why it was not chosen for verification.

Further insights into the models' discriminative capabilities can be derived from Table 5 and 6, while SDv2.1B starts with lower accuracy in distinguishing itself, a significant improvement in accuracy is seen across iterations. The initially anomalous behavior transitions into more discriminating regeneration.

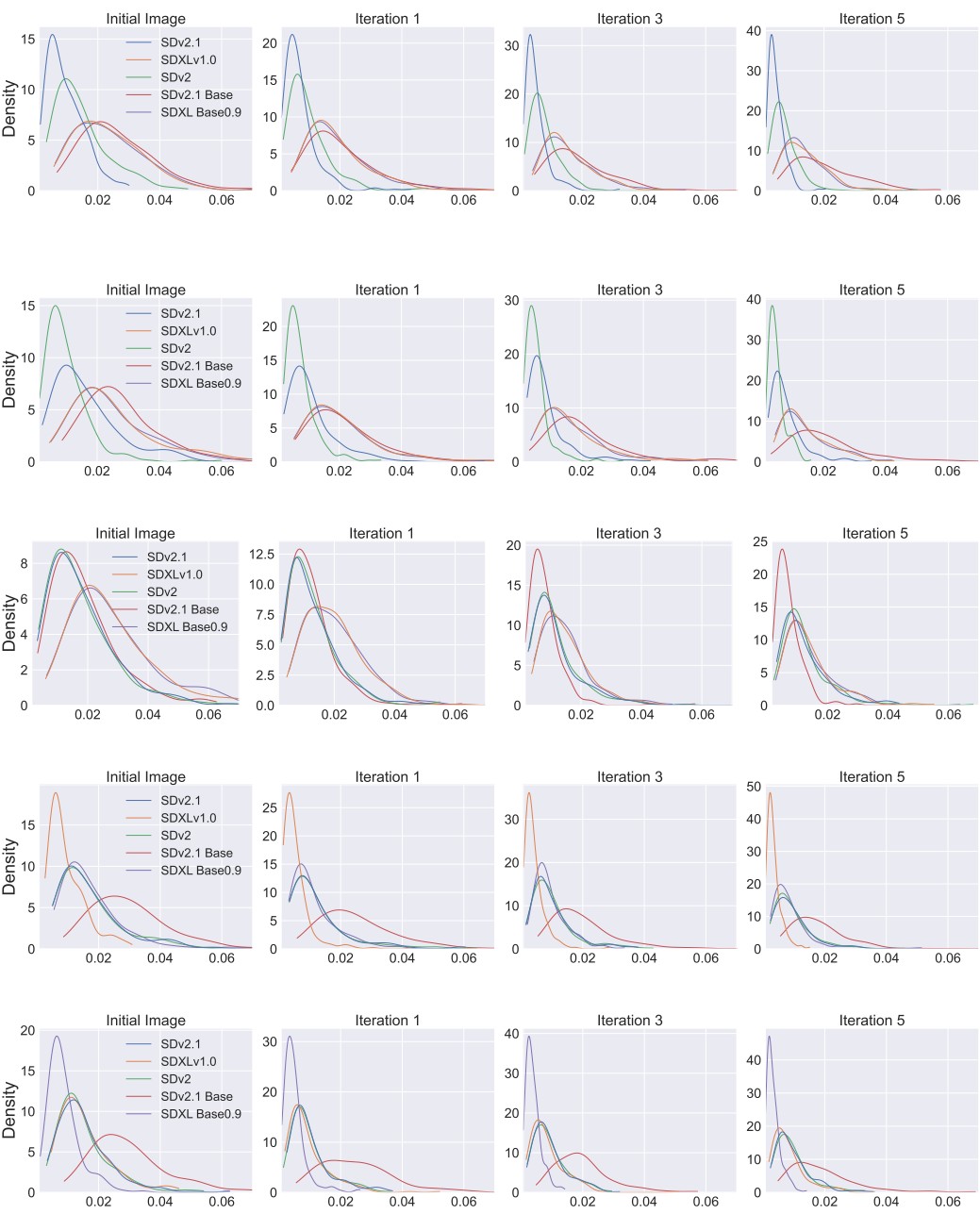

Figure 11: One Step Regeneration for various authentic models on the Polo Dataset using the CLIP metric at different iterations. The authentic models from top to bottom are: 1) SDv 2.1, 2) SD v2, 3) SD v2.1B,4) SDXL 1.0, 5) SDXL0.9B.

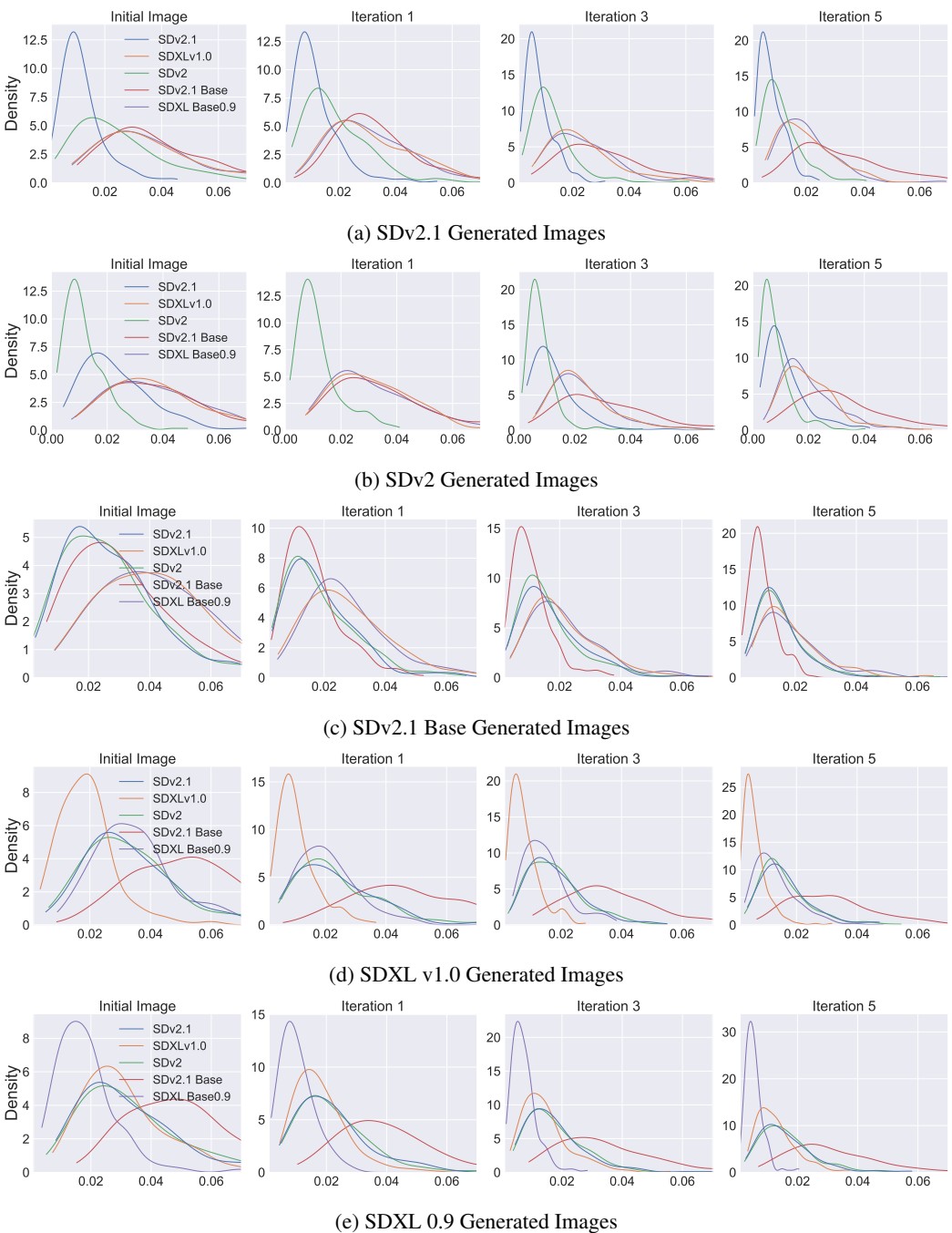

Figure 12: One Step Regeneration for various Authentic Models on the Coco Dataset using the CLIP metric at different iterations. The authentic models from top to bottom are: 1) SDv 2.1, 2) SD v2, 3) SD v2.1B, 4) SDXL 1.0, 5) SDXL0.9B.

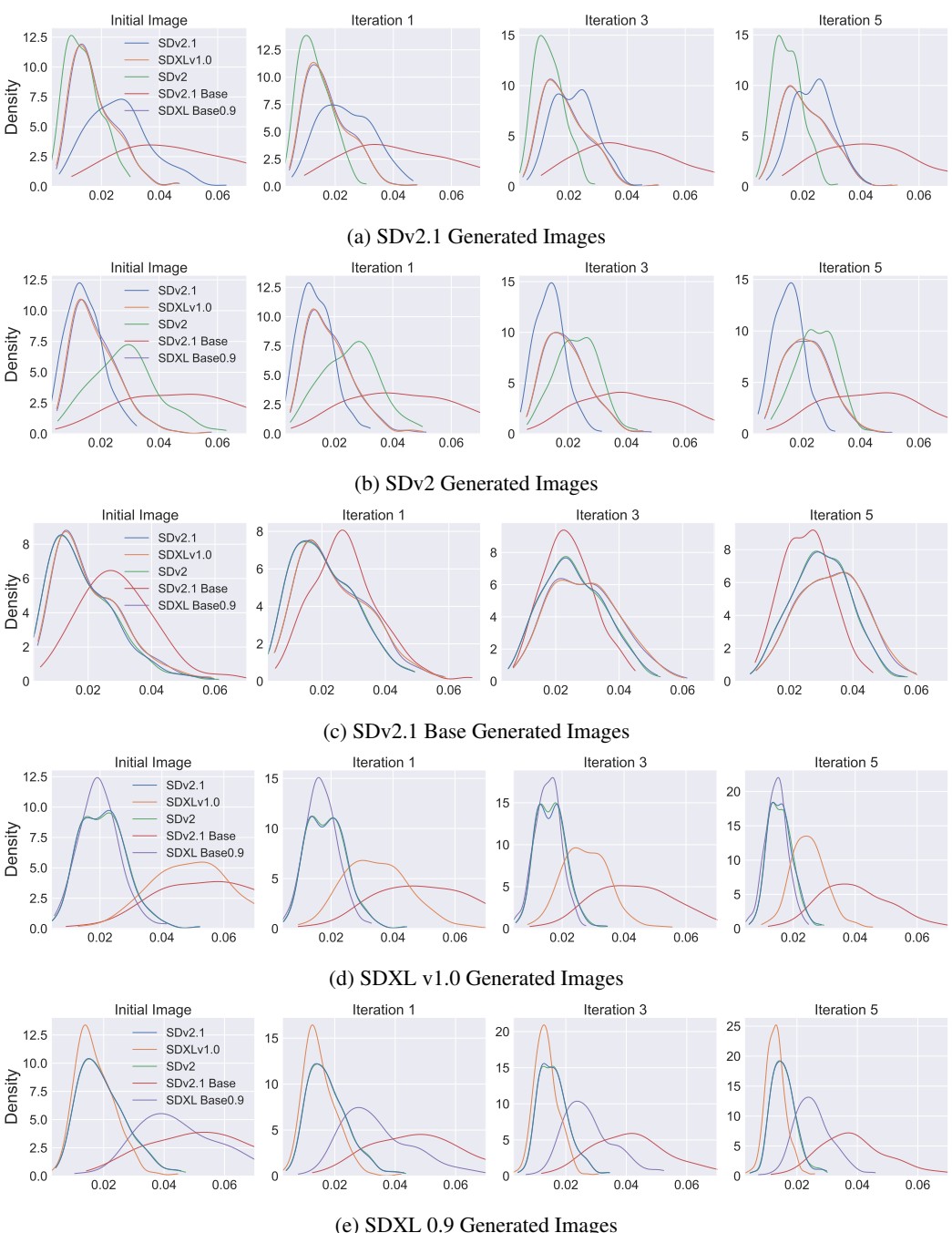

Figure 13: One Step Regeneration for various Authentic Models on the Polo Dataset using the LPIPS metric at different iterations. The authentic models from top to bottom are: 1) SDv 2.1, 2) SD v2, 3) SD v2.1B, 4) SDXL 1.0, 5) SDXL0.9B.

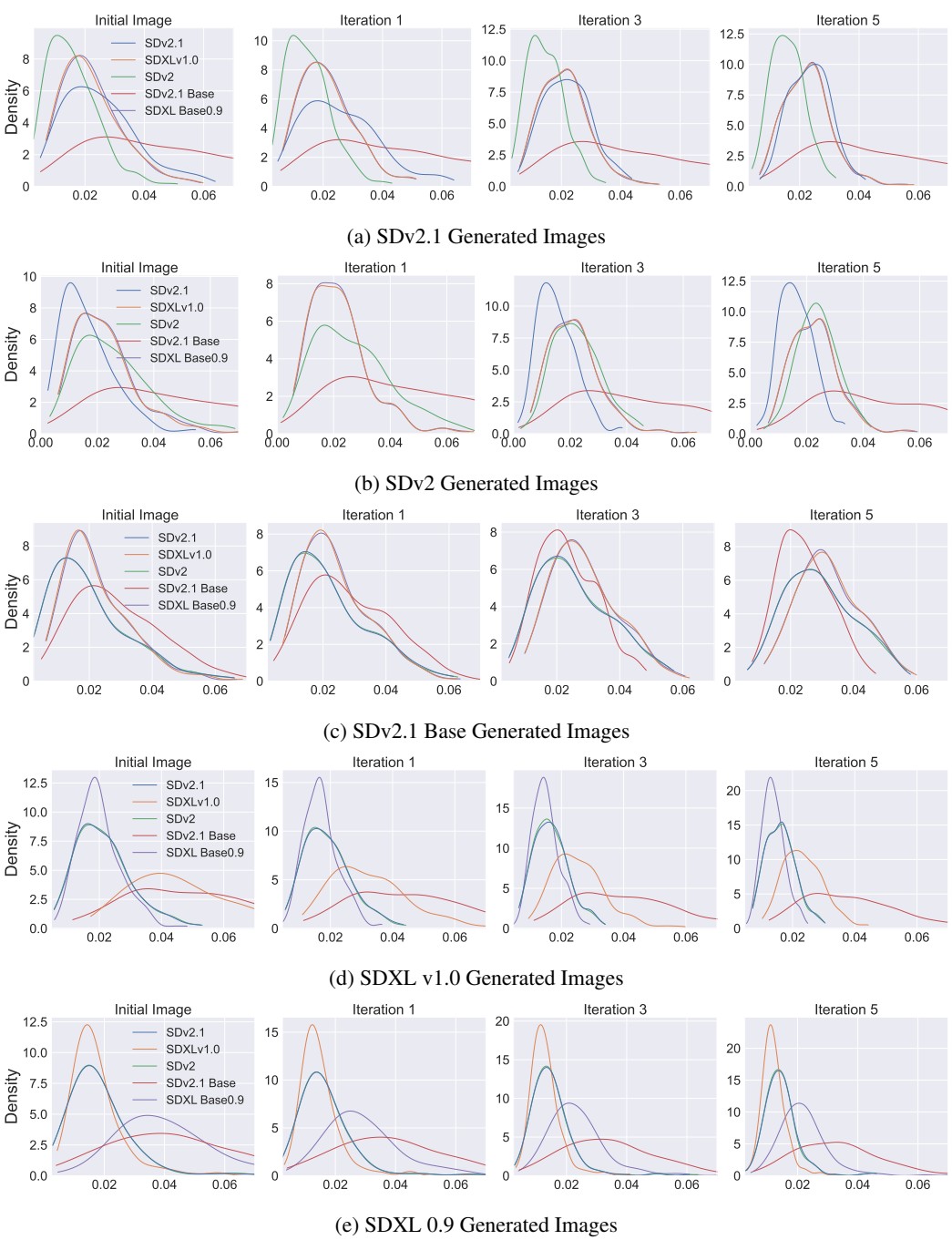

Figure 14: One Step Regeneration for various Authentic Models on the Coco Dataset using the LPIPS metric at different iterations. The authentic models from top to bottom are: 1) SDv 2.1, 2) SD v2, 3) SD v2.1B, 4) SDXL 1.0, 5) SDXL0.9B.

Table 5: The accuracy (Acc.) and misclassification rate (Mis.) of verifying the authentic models $\mathcal{G}_a$ using different contrast models $\mathcal{G}_\times$ on Coco Dataset.

| $\mathcal{G}_a$ \ $\mathcal{G}_\times$ | SD v2.1 | | SD v2. | | SD v2.1B | | SDXL 0.9 | | SDXL 1.0 | |
|---|---|---|---|---|---|---|---|---|---|---|
| | Acc. ↑ | Mis. ↓ | Acc. ↑ | Mis. ↓ | Acc. ↑ | Mis. ↓ | Acc. ↑ | Mis. ↓ | Acc. ↑ | Mis. ↓ |
| (a) $k = 1$ | | | | | | | | | | |
| **SD v2.1** | - | - | 87.5 | 9.5 | 97.5 | 2.5 | 96.5 | 2.0 | 97.0 | 2.5 |
| **SD v2.** | 89.0 | 8.5 | - | - | 97.0 | 2.0 | 96.0 | 3.0 | 95.5 | 2.5 |
| **SD v2.1B** | 14.5 | 81.0 | 2.0 | 97.0 | - | - | 47.5 | 47.0 | 48.5 | 44.0 |
| **SDXL 0.9** | 85.5 | 11.5 | 88.0 | 8.5 | 99.5 | 0.0 | - | - | 90.5 | 5.0 |
| **SDXL 1.0** | 92.5 | 4.0 | 89.5 | 7.0 | 98.0 | 1.5 | 95.5 | 3.0 | - | - |
| (b) $k = 3$ | | | | | | | | | | |
| **SD v2.1** | - | - | 73.0 | 23.0 | 97.5 | 1.5 | 94.5 | 3.5 | 95.5 | 3.5 |
| **SD v2.** | 72.0 | 21.0 | - | - | 97.0 | 1.0 | 95.0 | 4.0 | 94.5 | 4.5 |
| **SD v2.1B** | 74.0 | 21.0 | 74.5 | 20.5 | - | - | 82.0 | 14.0 | 81.5 | 14.5 |
| **SDXL 0.9** | 88.5 | 10.0 | 90.0 | 8.5 | 98.0 | 1.5 | - | - | 86.5 | 12.0 |
| **SDXL 1.0** | 91.5 | 5.5 | 92.5 | 5.0 | 100.0 | 0.0 | 89.0 | 9.0 | - | - |
| (c) $k = 5$ | | | | | | | | | | |
| **SD v2.1** | - | - | 66.0 | 27.0 | 99.5 | 0.5 | 94.5 | 3.0 | 93.0 | 4.0 |
| **SD v2.** | 72.5 | 18.5 | - | - | 96.0 | 3.5 | 91.0 | 7.5 | 94.0 | 5.0 |
| **SD v2.1B** | 78.5 | 16.5 | 77.0 | 17.5 | - | - | 82.5 | 13.0 | 85.5 | 10.5 |
| **SDXL 0.9** | 97.0 | 2.0 | 97.5 | 2.0 | 100.0 | 0.0 | - | - | 92.0 | 5.0 |
| **SDXL 1.0** | 93.0 | 4.0 | 92.5 | 7.0 | 99.0 | 0.5 | 88.0 | 9.5 | - | - |

Table 6: The accuracy (Acc.) and misclassification rate (Mis.) of verifying the authentic models $\mathcal{G}_a$ using different contrast models $\mathcal{G}_\times$ on Polo Dataset.

| $\mathcal{G}_a$ \ $\mathcal{G}_\times$ | SD v2.1 | | SD v2. | | SD v2.1B | | SDXL 0.9 | | SDXL 1.0 | |
|---|---|---|---|---|---|---|---|---|---|---|
| | Acc. ↑ | Mis. ↓ | Acc. ↑ | Mis. ↓ | Acc. ↑ | Mis. ↓ | Acc. ↑ | Mis. ↓ | Acc. ↑ | Mis. ↓ |
| (a) $k = 1$ | | | | | | | | | | |
| **SD v2.1** | - | - | 76.0 | 21.5 | 97.5 | 1.0 | 93.5 | 3.0 | 94.0 | 5.0 |
| **SD v2.** | 79.0 | 16.0 | - | - | 98.0 | 1.5 | 92.0 | 4.5 | 94.0 | 4.5 |
| **SD v2.1B** | 40.5 | 49.0 | 39.0 | 54.0 | - | - | 72.5 | 20.0 | 73.5 | 19.5 |
| **SDXL 0.9** | 86.0 | 11.5 | 85.0 | 10.5 | 99.5 | 0.5 | - | - | 91.5 | 4.5 |
| **SDXL 1.0** | 89.0 | 8.0 | 89.5 | 7.0 | 98.0 | 1.0 | 91.5 | 4.5 | - | - |
| (b) $k = 3$ | | | | | | | | | | |
| **SD v2.1** | - | - | 77.0 | 17.0 | 99.5 | 0.0 | 95.5 | 2.0 | 96.5 | 2.5 |
| **SD v2.** | 75.5 | 18.0 | - | - | 98.0 | 2.0 | 96.5 | 2.0 | 97.5 | 2.5 |
| **SD v2.1B** | 70.0 | 23.5 | 71.5 | 21.5 | - | - | 84.0 | 9.5 | 84.5 | 10.5 |
| **SDXL 0.9** | 92.5 | 5.0 | 91.0 | 6.0 | 100.0 | 0.0 | - | - | 91.0 | 6.0 |
| **SDXL 1.0** | 90.0 | 7.5 | 89.5 | 8.5 | 100.0 | 0.0 | 90.0 | 7.0 | - | - |
| (c) $k = 5$ | | | | | | | | | | |
| **SD v2.1** | - | - | 80.0 | 14.0 | 99.5 | 0.5 | 98.5 | 0.5 | 99.0 | 0.0 |
| **SD v2.** | 77.5 | 18.5 | - | - | 99.0 | 0.0 | 96.5 | 3.0 | 95.5 | 2.5 |
| **SD v2.1B** | 81.5 | 16.5 | 82.5 | 13.5 | - | - | 88.5 | 8.0 | 89.5 | 9.0 |
| **SDXL 0.9** | 97.5 | 2.5 | 96.5 | 2.0 | 99.5 | 0.5 | - | - | 92.0 | 6.5 |
| **SDXL 1.0** | 95.5 | 4.0 | 92.5 | 4.5 | 100.0 | 0.0 | 94.5 | 4.0 | - | - |

## C    MODEL SETTINGS AND METRICS OF TEXT GENERATION

To mitigate biases stemming from varied sampling strategies, we set the temperature and top p to 0.7 and 0.95 for all models and experiments. For prompt-based experiments, we use the following prompts:

- Translate to English: *You are a professional translator. You should translate the following sentence to English and output the final result only:* {***INPUT***}

- Translate to French: *You are a professional translator. You should translate the following sentence to French and output the final result only:* {***INPUT***}

In our text generation experiments, we consider the following metrics:

- **BLEU**: calculate the precision of the overlap of various n-grams (usually 1 to 4) between the candidate and the reference (Papineni et al., 2002).

- **ROUGE-L**: calculate the F1 of longest common subsequence between the candidate and reference (Lin, 2004).

- **BERTScore**: compute the similarity between candidate and reference tokens using cosine similarity on their embeddings. Then, the token-level scores are aggregated to produce a single score for the whole text (Zhang et al., 2020).

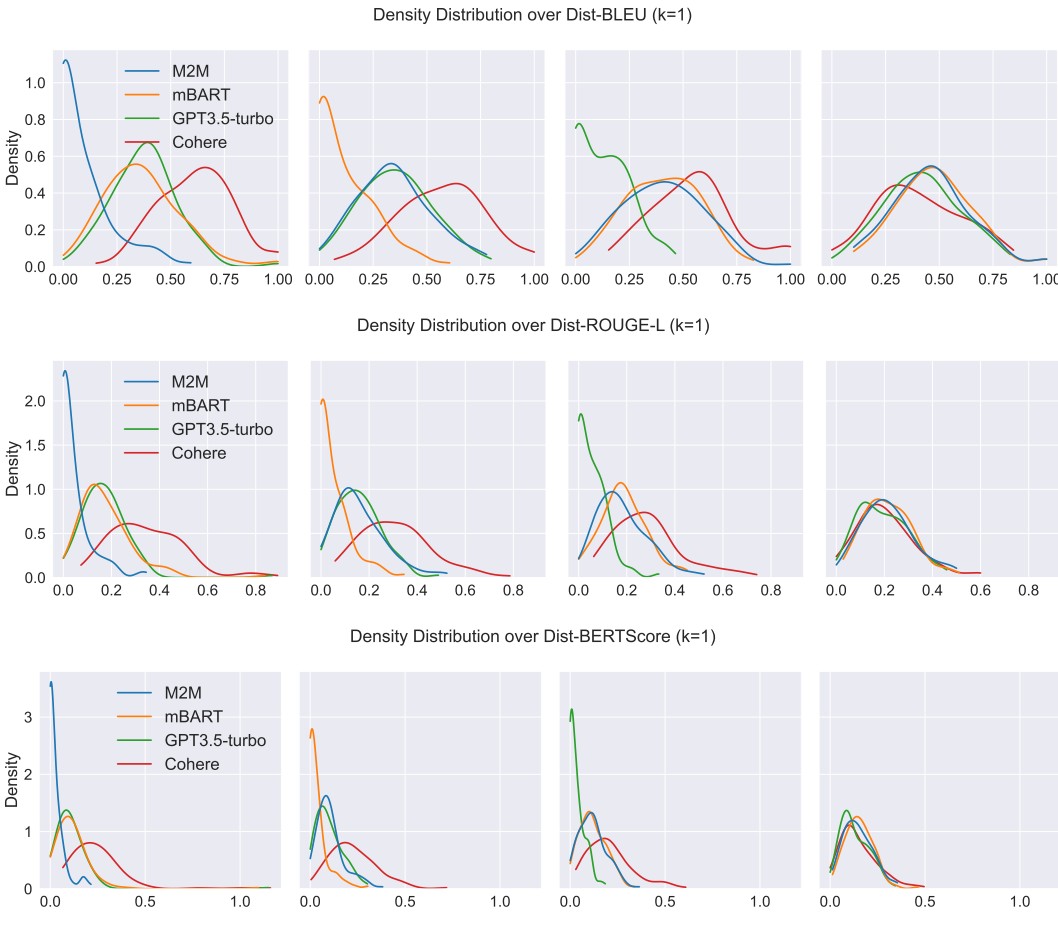

Figure 15: Density distribution of one-step regeneration among four text generation models, where the input to the one-step regeneration is the first iteration from the authentic models. The authentic models from left to right are: 1) M2M, 2) mBART, 3) GPT3.5-turbo, and 4) Cohere.

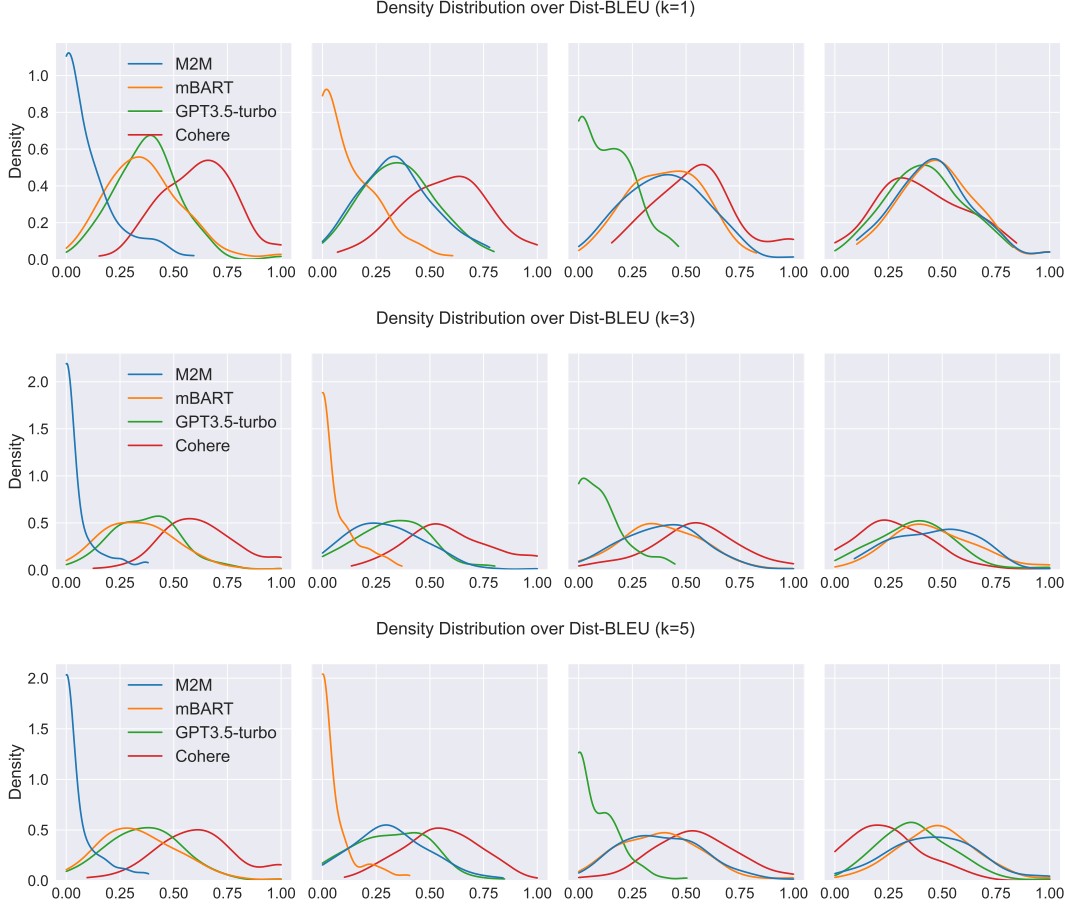

Figure 16: Density distribution of one-step regeneration among four text generation models, where the input to the one-step regeneration is the $k$th iteration from the authentic models. The authentic models from left to right are: 1) M2M, 2) mBART, 3) GPT3.5-turbo, and 4) Cohere.

## D  SUPPLEMENTARY EXPERIMENTS FOR TEXT GENERATION MODELS

### D.1  DENSITY DISTRIBUTION OF A ONE-STEP REGENERATION

In Figure 15, we illustrate the density distribution of one-step regeneration for four text generation models, using the first iteration from the authentic models as input. Excluding Cohere, the density distributions of the authentic models are discernible from those of the contrast models across BLEU, ROUGE-L, and BERTScore metrics.

Figure 16 illustrates the density distribution when the input is the $k$th iteration from the authentic models. As $k$ increases, the distinction between the authentic model's distribution and that of the contrast models becomes more pronounced.

### D.2  EXPERIMENTS FOR GPT4

In this section, we analyze the characteristics of GPT4. We first present the density distribution when the input is the $k$th iteration from the authentic models in Figure 17. Similarly, as $k$ increases, the difference between the authentic model's distribution and the contrast models intensifies. Nonetheless, distinguishing between GPT3.5-turbo and GPT4 proves difficult, particularly when GPT3.5-turbo serves as the authentic model, even for larger values of $k$. Detailed examination reveals that GPT4's one-step regeneration bears resemblance to the $k$-th iteration of GPT3.5-turbo. This might stem from GPT3.5-turbo and GPT4 originating from the same institution, suggesting potential similarities in architecture and pre-training data. Thus, we contend that models stemming from the same institution

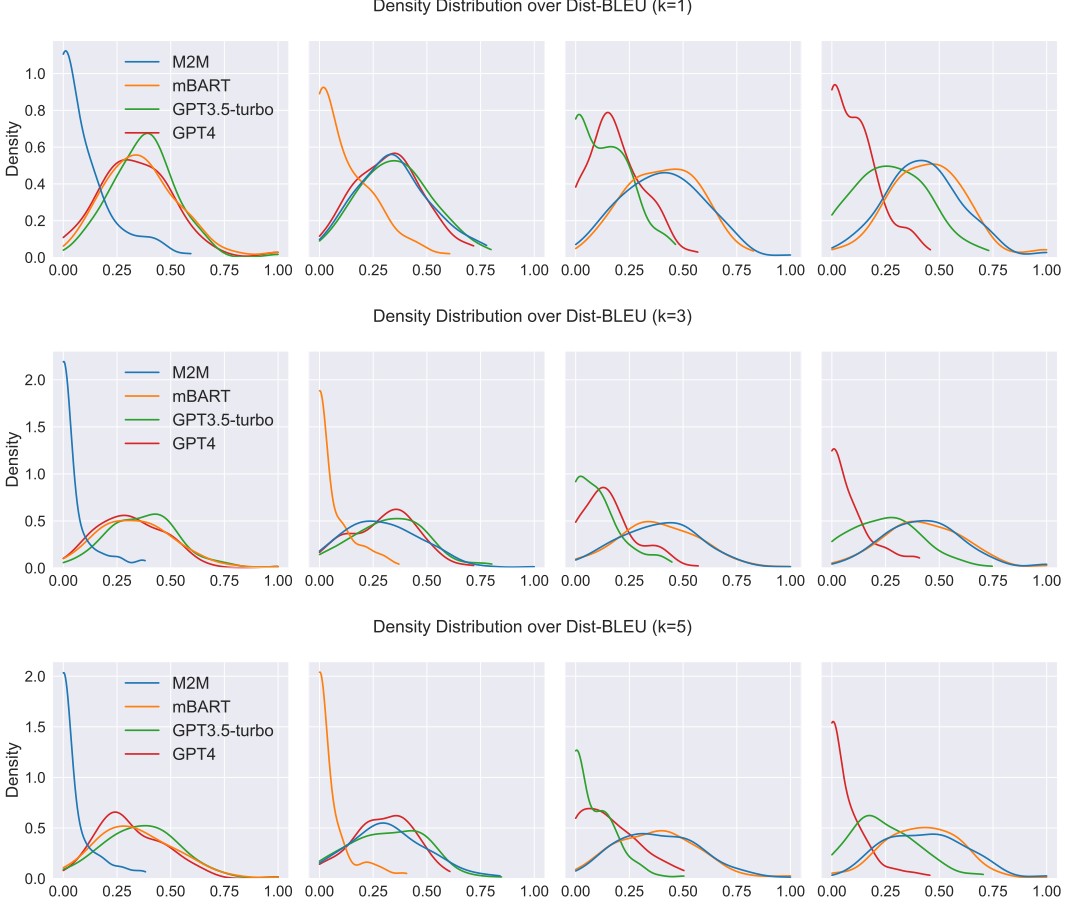

Figure 17: Density distribution of one-step regeneration among four text generation models, where the input to the one-step regeneration is the $k$th iteration from the authentic models. The authentic models from left to right are: 1) M2M, 2) mBART, 3) GPT3.5-turbo, and 4) GPT4.

inherently share identical intellectual property, thus obviating the possibility of intellectual property conflicts. Interestingly, when GPT4 is the authentic model, our methodology can differentiate it from GPT3.5-turbo, attributed to the marked difference between GPT3.5-turbo's one-step regeneration and GPT4's $k$-th iteration. This distinction, we surmise, is due to GPT4's superior advancement over GPT3.5-turbo.

In evaluating the verification of authentic models, our methodology, as depicted in Table 7, consistently confirms the authorship of all models with an accuracy exceeding 85%. Furthermore, the misclassification rate remains below 10%.

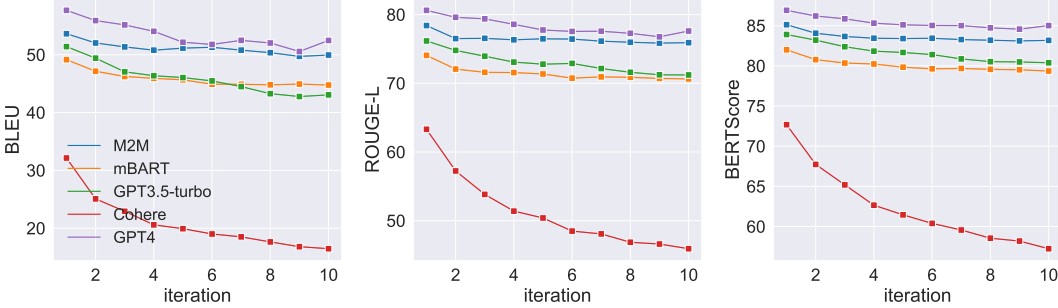

Figure 18: The quality between the original inputs and $k$th regeneration using BLEU, ROULGE-L and BERTScore.

Table 7: The accuracy (Acc.) and misclassification rate (Mis.) of verifying the authentic models ($\mathcal{G}_a$) using different contrast models ($\mathcal{G}_\times$).

| $\mathcal{G}_\times$ / $\mathcal{G}_a$ | M2M | | mBART | | GPT3.5-turbo | | GPT4 | |
|---|---|---|---|---|---|---|---|---|
| | Acc. ↑ | Mis. ↓ | Acc. ↑ | Mis. ↓ | Acc. ↑ | Mis. ↓ | Acc. ↑ | Mis. ↓ |
| (a) $k = 1$ | | | | | | | | |
| M2M | - | - | 94.0 | 2.0 | 93.0 | 5.0 | 87.0 | 8.0 |
| mBART | 85.0 | 11.0 | - | - | 89.0 | 8.0 | 80.0 | 15.0 |
| GPT3.5-turbo | 87.0 | 8.0 | 90.0 | 8.0 | - | - | 51.0 | 31.0 |
| GPT4 | 92.0 | 2.0 | 94.0 | 2.0 | 77.0 | 10.0 | - | - |
| (b) $k = 3$ | | | | | | | | |
| M2M | - | - | 94.0 | 1.0 | 95.0 | 3.0 | 94.0 | 5.0 |
| mBART | 85.0 | 7.0 | - | - | 90.0 | 4.0 | 89.0 | 5.0 |
| GPT3.5-turbo | 89.0 | 8.0 | 93.0 | 5.0 | - | - | 51.0 | 28.0 |
| GPT4 | 96.0 | 3.0 | 92.0 | 6.0 | 79.0 | 9.0 | - | - |
| (c) $k = 5$ | | | | | | | | |
| M2M | - | - | 94.0 | 2.0 | 95.0 | 1.0 | 94.0 | 4.0 |
| mBART | 91.0 | 4.0 | - | - | 88.0 | 6.0 | 89.0 | 6.0 |
| GPT3.5-turbo | 90.0 | 6.0 | 94.0 | 2.0 | - | - | 55.0 | 17.0 |
| GPT4 | 97.0 | 2.0 | 94.0 | 3.0 | 83.0 | 5.0 | - | - |

## D.3 QUALITY OF RE-GENERATED SENTENCES AND IMAGES

### D.3.1 TEXT REGENERATIONS

In this section, we examine the quality of re-generated sentences. As illustrated in Figure 18, the re-generated sentences display high quality across most evaluation metrics, except for Cohere. The slower convergence observed for Cohere can be attributed to its subpar re-generation quality, which significantly deviates from that of other models. For further illustration, refer to the samples and their regenerations in Table 8-11.

### D.3.2 IMAGE REGENERATIONS

An integral part of our evaluation process involves understanding the reproducibility of the images generated by different models across iterations. To this end, we present image re-generations for various authentic models in Figure 19-23. These re-generations provide insights into the consistency and stability of each model when tasked with reproducing the same visual content over multiple iterations.

Table 8: Three original samples and their corresponding $k$-iteration from M2M.

| |
|---|
| Sample 1: |

**Original input**: Working in groups of 2 or 4 (even numbers may work better later on), suggest a heuristic to evaluate the utility of a given board configuration.
**Iteration 1**: Working in groups of 2 or 4 (even numbers can work better later), suggests a heurist to evaluate the usefulness of a given table configuration.
**Iteration 2**: Working in groups of 2 or 4 (even numbers can work better later), suggests a heurist to evaluate the usefulness of a given table configuration.
**Iteration 3**: Working in groups of 2 or 4 (even numbers can work better later), suggests a heurist to evaluate the usefulness of a given table configuration.
**Iteration 4**: Working in groups of 2 or 4 (even numbers can work better later), suggests a heurist to evaluate the usefulness of a given table configuration.
**Iteration 5**: Working in groups of 2 or 4 (even numbers can work better later), suggests a heurist to evaluate the usefulness of a given table configuration.

| |
|---|
| Sample 2: |

**Original input**: Therefore, to improve the ethical coping ability of Chinese NLP techniques and their application effectiveness in the field of ethics, the Chinese ethics knowledge base and KEPTMs for ethics domain are researched
**Iteration 1**: Therefore, in order to improve the ethical treatment capacity of Chinese NLP techniques and their effectiveness of application in the field of ethics, the knowledge base of Chinese ethics and the KEPTMs for the field of ethics are studied.
**Iteration 2**: Therefore, in order to improve the ethical treatment capacity of Chinese NLP techniques and their effectiveness of application in the field of ethics, the knowledge base of Chinese ethics and KEPTMs for the field of ethics are studied.
**Iteration 3**: Therefore, in order to improve the ethical treatment capacity of Chinese NLP techniques and their effectiveness of application in the field of ethics, the knowledge base of Chinese ethics and KEPTMs for the field of ethics are studied.
**Iteration 4**: Therefore, in order to improve the ethical treatment capacity of Chinese NLP techniques and their effectiveness of application in the field of ethics, the knowledge base of Chinese ethics and KEPTMs for the field of ethics are studied.
**Iteration 5**: Therefore, in order to improve the ethical treatment capacity of Chinese NLP techniques and their effectiveness of application in the field of ethics, the knowledge base of Chinese ethics and KEPTMs for the field of ethics are studied.

| |
|---|
| Sample 3: |

**Original input**: I kindly request a modification to the due date for my assignment. According to the previous thread, I was informed that I would receive an extension of 33 hours for this assignment. However, as of now, my Moodle account still indicates the original due date
**Iteration 1**: I slowly ask for a change to the appropriate date for my task. According to the previous topic, I was informed that I would receive an extension of 33 hours for this task. However, from now on, my Moodle account still indicates the appropriate original date.
**Iteration 2**: I slowly ask for a change to the appropriate date for my task. According to the previous subject, I was informed that I would receive a 33-hour extension for this task. However, from now on, my Moodle account still indicates the appropriate original date.
**Iteration 3**: I slowly ask for a change to the appropriate date for my task. According to the previous topic, I was informed that I would receive a 33-hour extension for this task. However, from now on, my Moodle account still indicates the appropriate original date.
**Iteration 4**: I slowly ask for a change to the appropriate date for my task. According to the previous topic, I was informed that I would receive a 33-hour extension for this task. However, from now on, my Moodle account still indicates the appropriate original date.
**Iteration 5**: I slowly ask for a change to the appropriate date for my task. According to the previous topic, I was informed that I would receive an extension of 33 hours for this task. However, from now on, my Moodle account still indicates the appropriate original date.

Table 9: Three original samples and their corresponding $k$-iteration from mBART.

| Sample 1: |
| --- |

**Original input**: Working in groups of 2 or 4 (even numbers may work better later on), suggest a heuristic to evaluate the utility of a given board configuration.
**Iteration 1**: If you work in groups of 2 or 4 (even numbers can work better later), suggest a mouse kit to evaluate the usefulness of a given configuration of the table.
**Iteration 2**: If you work in groups of 2 or 4 (even numbers can work better later), suggest a mouse kit to evaluate the usefulness of a particular table configuration.
**Iteration 3**: If you work in groups of 2 or 4 (even numbers can work better later), suggest a mouse kit to evaluate the usefulness of a particular table configuration.
**Iteration 4**: If you work in groups of 2 or 4 (even numbers can work better later), suggest a mouse kit to evaluate the usefulness of a given table configuration.
**Iteration 5**: If you work in groups of 2 or 4 (even numbers can work better later), suggest a mouse kit to evaluate the usefulness of a given table configuration.

| Sample 2: |
| --- |

**Original input**: Therefore, to improve the ethical coping ability of Chinese NLP techniques and their application effectiveness in the field of ethics, the Chinese ethics knowledge base and KEPTMs for ethics domain are researched
**Iteration 1**: For this reason, to improve the ethical adaptation capacity of Chinese NLP techniques and their effective application in the ethics field, the Chinese Ethics Knowledge Base and the KEPTMs for the ethics field are being studied.
**Iteration 2**: Therefore, to improve the ethical adaptation capacity of Chinese NLP techniques and their effective application in the field of ethics, the Chinese ethics knowledge base and the ECEs for the field of ethics are being studied.
**Iteration 3**: Consequently, in order to improve the ethical adaptation capacity of Chinese PLN techniques and their effective application in the field of ethics, the Chinese ethics knowledge base and ECE for the field of ethics are being studied.
**Iteration 4**: Consequently, in order to improve the ethical adaptation capacity of Chinese PLN techniques and their effective application in the field of ethics, the Chinese ethics knowledge base and UNECE for the field of ethics are being studied.
**Iteration 5**: Consequently, in order to improve the ethical adaptation capacity of Chinese PLN techniques and their effective application in the field of ethics, the Chinese ethics knowledge base and UNECE for the field of ethics are being studied.

| Sample 3: |
| --- |

**Original input**: I kindly request a modification to the due date for my assignment. According to the previous thread, I was informed that I would receive an extension of 33 hours for this assignment. However, as of now, my Moodle account still indicates the original due date.
**Iteration 1**: I am requesting a change to the due date of my assignment, and according to the previous thread, I have been informed that I will receive a 33-hour extension for this assignment, but at the moment my Moodle account still indicates the original due date.
**Iteration 2**: I am requesting a change to my assignment due date and, according to the previous thread, I have been informed that I have received a 33-hour extension for this assignment, but at the moment, my Moodle account still indicates the original due date.
**Iteration 3**: I am requesting a change to my assignment due date and, according to the previous thread, I have been informed that I have received a 33-hour extension for this assignment, but for the time being, my Moodle account still indicates the original due date.
**Iteration 4**: I am requesting a change to the due date of my assignment and, according to the previous thread, I have been informed that I have received a 33-hour extension for this assignment, but for the time being, my Moodle account still indicates the original due date.
**Iteration 5**: I am requesting a change to the due date of my assignment and, according to the previous thread, I was informed that I have received a 33-hour extension for this assignment, but for the time being, my Moodle account still indicates the original due date.

Table 10: Three original samples and their corresponding $k$-iteration from GPT3.5-turbo.

| Sample 1: |
|---|

**Original input**: Working in groups of 2 or 4 (even numbers may work better later on), suggest a heuristic to evaluate the utility of a given board configuration.
**Iteration 1**: Working in groups of 2 or 4 (even numbers might work better later on), propose a heuristic to evaluate the usefulness of a given board configuration.
**Iteration 2**: Working in groups of 2 or 4 (even numbers might work better in the future), propose a heuristic to evaluate the usefulness of a given board configuration.
**Iteration 3**: Working in groups of 2 or 4 (even numbers might work better in the future), propose a heuristic to evaluate the utility of a given board configuration.
**Iteration 4**: Working in groups of 2 or 4 (even numbers might work better in the future), propose a heuristic to evaluate the usefulness of a given board configuration.
**Iteration 5**: Working in groups of 2 or 4 (even numbers might work better in the future), propose a heuristic to evaluate the usefulness of a given board configuration.

| Sample 2: |
|---|

**Original input**: Therefore, to improve the ethical coping ability of Chinese NLP techniques and their application effectiveness in the field of ethics, the Chinese ethics knowledge base and KEPTMs for ethics domain are researched
**Iteration 1**: Therefore, in order to improve the ethical adaptability of Chinese NLP techniques and their application efficiency in the field of ethics, the Chinese ethics knowledge base and KEPTMs for the field of ethics are studied.
**Iteration 2**: Therefore, in order to improve the ethical adaptability of Chinese natural language processing techniques and their effectiveness in the field of ethics, Chinese ethical knowledge base and KEPTMs for the field of ethics are studied.
**Iteration 3**: Therefore, in order to improve the ethical adaptability of Chinese natural language processing techniques and their effectiveness in the field of ethics, Chinese ethical knowledge base and KEPTMs for the field of ethics are studied.
**Iteration 4**: Therefore, in order to improve the ethical adaptability of Chinese natural language processing techniques and their effectiveness in the field of ethics, Chinese ethical knowledge base and KEPTMs for the field of ethics are being studied.
**Iteration 5**: Therefore, in order to improve the ethical adaptability of Chinese natural language processing techniques and their effectiveness in the field of ethics, Chinese ethical knowledge base and KEPTMs for the field of ethics are being studied.

| Sample 3: |
|---|

**Original input**: I kindly request a modification to the due date for my assignment. According to the previous thread, I was informed that I would receive an extension of 33 hours for this assignment. However, as of now, my Moodle account still indicates the original due date.
**Iteration 1**: I politely request a modification of the due date for my assignment. According to the previous thread, I was informed that I would be granted a 33-hour extension for this assignment. However, at the moment, my Moodle account still shows the original due date.
**Iteration 2**: I politely request a modification of the deadline for my assignment. According to the previous thread, I was informed that I would be granted an additional 33 hours for this assignment. However, at the moment, my Moodle account still displays the original deadline.
**Iteration 3**: I politely request a change in the deadline for my assignment. According to the previous thread, I was informed that I would have an additional 33 hours for this assignment. However, currently, my Moodle account still displays the original deadline.
**Iteration 4**: I politely request a change in deadline for my assignment. According to the previous thread, I was informed that I would have an additional 33 hours for this assignment. However, currently, my Moodle account still displays the original deadline.
**Iteration 5**: I kindly request a change of deadline for my assignment. According to the previous thread, I was informed that I would have an additional 33 hours for this assignment. However, currently, my Moodle account still displays the original deadline.

Table 11: Three original samples and their corresponding $k$-iteration from Cohere.

| |
|---|
| Sample 1: |

**Original input**: Working in groups of 2 or 4 (even numbers may work better later on), suggest a heuristic to evaluate the utility of a given board configuration.
**Iteration 1**: Working in groups of 2 or 4 (odd numbers may work better later), suggests a heuristic to evaluate the utility of a seating chart configuration.
**Iteration 2**: Working in groups of 2 or 4 (odd numbers may work better later), suggests a heuristic for evaluating the utility of a configuration of the seating plan.
**Iteration 3**: Working in groups of 2 or 4 (odd numbers can work better later), suggests an algorithm for evaluating the utility of a assistance plan configuration.
**Iteration 4**: Working in groups of 2 or 4 (odd numbers can work better later), suggests an algorithm to evaluate the utility of a plan of assistance configuration.
**Iteration 5**: Working in groups of 2 or 4 (odd numbers may work better later), suggests an algorithm to evaluate the utility of a configuration of assistance plans.

| |
|---|
| Sample 2: |

**Original input**: Therefore, to improve the ethical coping ability of Chinese NLP techniques and their application effectiveness in the field of ethics, the Chinese ethics knowledge base and KEPTMs for ethics domain are researched
**Iteration 1**: Therefore, to improve the ethical sweeping capacity of Chinese NLP techniques and their effectiveness in the ethical domain, Chinese knowledge bases on ethics and KEPTMs for the ethical domain are sought.
**Iteration 2**: Therefore, to improve the ethical brushing capacity of Chinese NLP techniques and their effectiveness in the ethical domain, Chinese knowledge bases on ethics and KEPTMs for the ethical domain are sought.
**Iteration 3**: Therefore, to improve the brushing capacity of Chinese NLP methods and their effectiveness in the ethical domain, Chinese knowledge bases on ethics and KEPTMs for the ethical domain are sought.
**Iteration 4**: Therefore, to improve the brushing capacity of Chinese NLP methods and their effectiveness in the ethical domain, Chinese knowledge bases on ethics and KEPTMs for the ethical domain are sought.
**Iteration 5**: Therefore, to improve the brushing capacity of Chinese NLP methods and their effectiveness in the ethical domain, Chinese knowledge bases on ethics and KEPTMs for the ethical domain are sought.

| |
|---|
| Sample 3: |

**Original input**: I kindly request a modification to the due date for my assignment. According to the previous thread, I was informed that I would receive an extension of 33 hours for this assignment. However, as of now, my Moodle account still indicates the original due date.
**Iteration 1**: Please extend the deadline for my assignment. I was informed that I will receive a 33 hour extension for this assignment, however, it still shows the initial date on my Moodle account.
**Iteration 2**: Please extend the deadline for my assignment. I was informed that I will receive a 33 hour extension for this assignment, or this amount is still visible on my Moodle account.
**Iteration 3**: Please extend the deadline for my assignment. I was informed that I will receive a 33 hour extension for this assignment, this quantity is always visible on my Moodle account.
**Iteration 4**: Would you like to request an extension of the deadline for my mission? I am informed that I will obtain a 33 hour extension for this mission, this quantity is still visible on my Moodle account.
**Iteration 5**: Did you want to request an extension of your mission deadline? I was informed that I will obtain a 33 hour extension for this mission, this amount is visible on my Moodle account.

Table 12: Three original samples and their corresponding $k$-iteration from GPT4.

| Sample 1: |
| --- |

**Original input**: Working in groups of 2 or 4 (even numbers may work better later on), suggest a heuristic to evaluate the utility of a given board configuration.
**Iteration 1**: Working in groups of 2 or 4 (even numbers may work better later), propose a heuristic to evaluate the usefulness of a given board configuration.
**Iteration 2**: Working in groups of 2 or 4 (even numbers may work better later), propose a heuristic to evaluate the usefulness of a given board configuration.
**Iteration 3**: Working in groups of 2 or 4 (even numbers may work better later), propose a heuristic to evaluate the usefulness of a given board configuration.
**Iteration 4**: Working in groups of 2 or 4 (even numbers may work better later), propose a heuristic to assess the usefulness of a given board configuration.
**Iteration 5**: Working in groups of 2 or 4 (even numbers may work better later), propose a heuristic to evaluate the usefulness of a given board configuration.

| Sample 2: |
| --- |

**Original input**: Therefore, to improve the ethical coping ability of Chinese NLP techniques and their application effectiveness in the field of ethics, the Chinese ethics knowledge base and KEPTMs for ethics domain are researched
**Iteration 1**: Therefore, to improve the ethical management capacity of Chinese NLP techniques and their application effectiveness in the field of ethics, the Chinese ethical knowledge base and KEPTMs for the field of ethics are being studied.
**Iteration 2**: Therefore, to improve the ethical management capacity of Chinese NLP techniques and their application effectiveness in the field of ethics, the Chinese ethical knowledge base and KEPTMs for the field of ethics are under study.
**Iteration 3**: Therefore, to improve the ethical management capacity of Chinese NLP techniques and their application effectiveness in the field of ethics, the Chinese ethical knowledge base and KEPTMs for the field of ethics are currently under study.
**Iteration 4**: Therefore, to improve the ethical management capacity of Chinese NLP techniques and their application efficiency in the field of ethics, the Chinese ethical knowledge base and KEPTMs for the field of ethics are currently under study.
**Iteration 5**: Therefore, to improve the ethical management capacity of Chinese NLP techniques and their application efficiency in the field of ethics, the Chinese ethical knowledge base and KEPTMs for the field of ethics are currently under study.

| Sample 3: |
| --- |

**Original input**: I kindly request a modification to the due date for my assignment. According to the previous thread, I was informed that I would receive an extension of 33 hours for this assignment. However, as of now, my Moodle account still indicates the original due date.
**Iteration 1**: I kindly request a modification of the due date for my assignment. According to the previous thread, I was informed that I would receive a 33-hour extension for this assignment. However, at the present time, my Moodle account still shows the initial due date.
**Iteration 2**: I kindly request a change in the due date for my assignment. According to the previous thread, I was informed that I would receive a 33-hour extension for this assignment. However, at present, my Moodle account still displays the initial due date.
**Iteration 3**: I kindly request a change of the due date for my assignment. According to the previous thread, I was informed that I would receive a 33-hour extension for this assignment. However, at present, my Moodle account still displays the initial due date.
**Iteration 4**: I am kindly requesting a change in the due date for my assignment. According to the previous thread, I was informed that I would receive a 33-hour extension for this task. However, at the present time, my Moodle account still displays the initial due date.
**Iteration 5**: I kindly request a change of the due date for my assignment. According to the previous thread, I was informed that I would receive a 33-hour extension for this task. However, at the present time, my Moodle account still displays the initial due date.

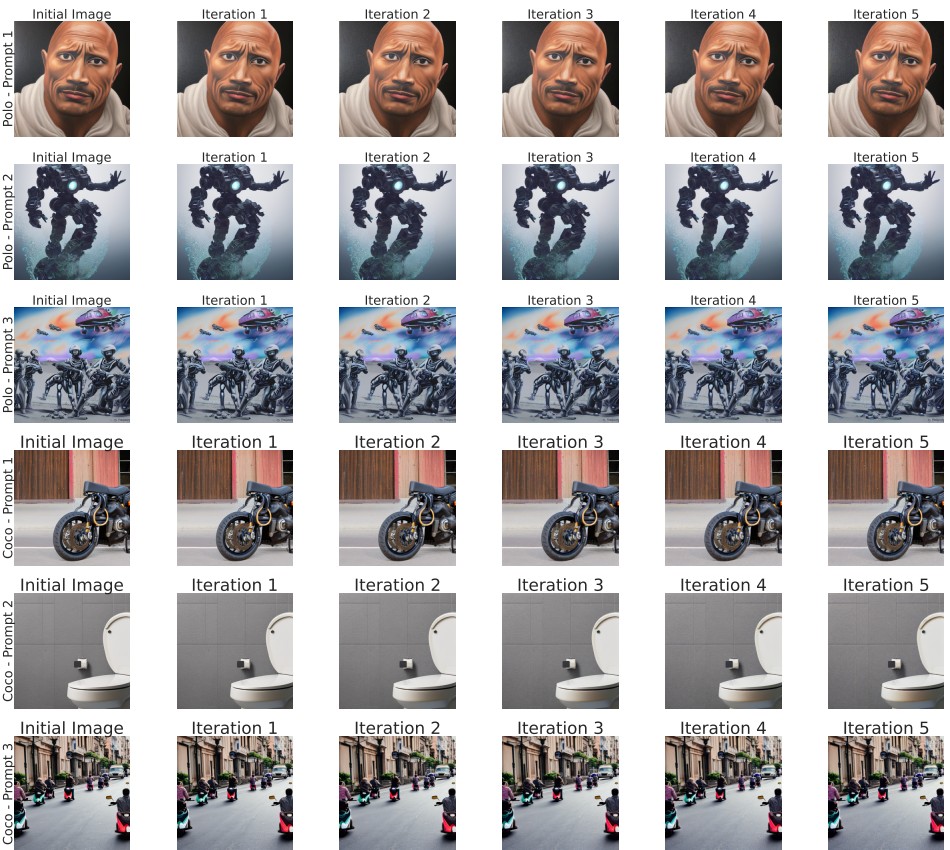

Figure 19: Initial image generation and subsequent regenerations by the SDv2.1 model on Coco and Polo datasets.

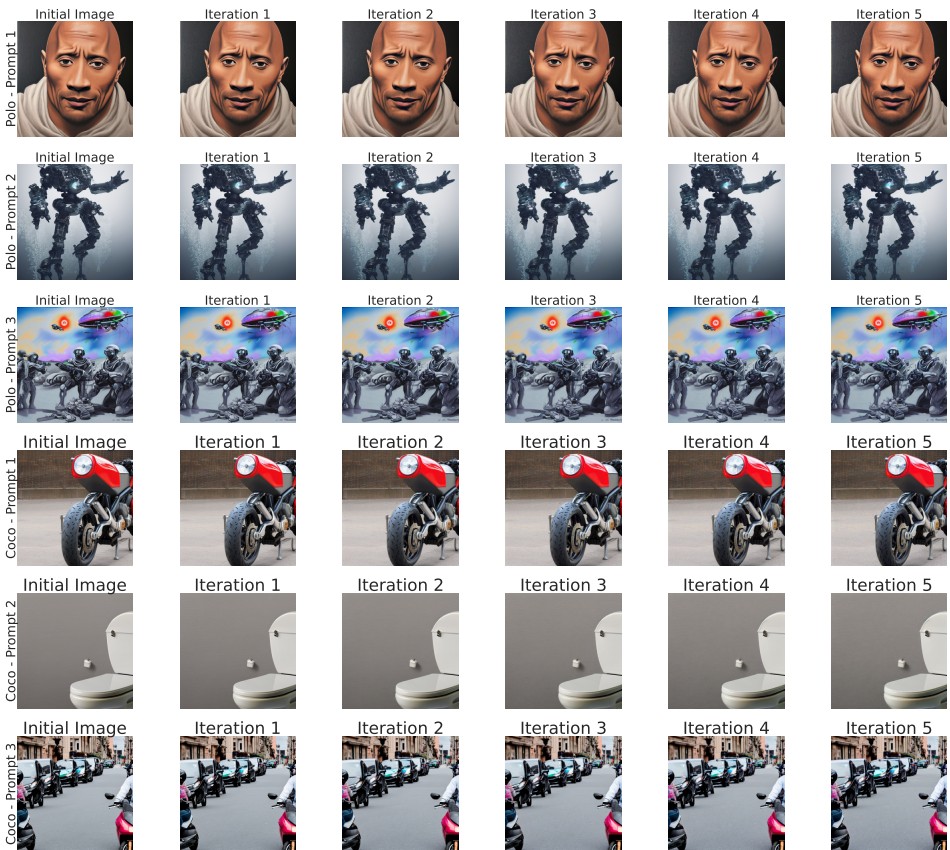

Figure 20: Initial image generation and subsequent regenerations by the SDv2 model on Coco and Polo datasets.

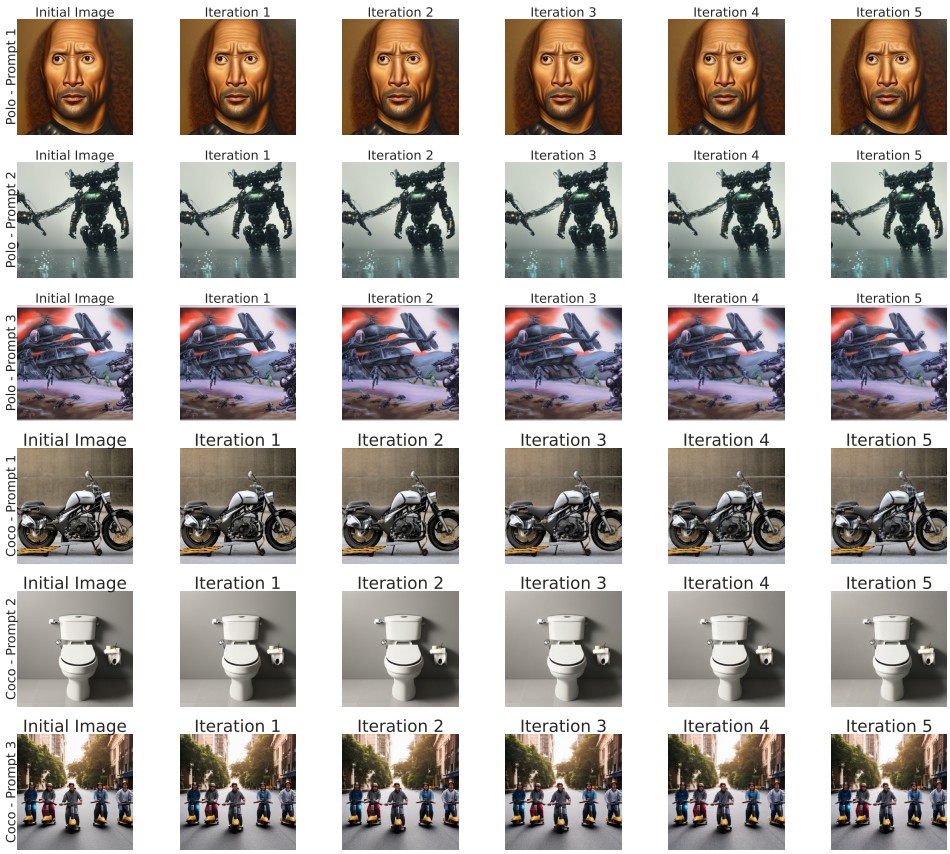

Figure 21: Initial image generation and subsequent regenerations by the SDv2.1B model on Coco and Polo datasets.

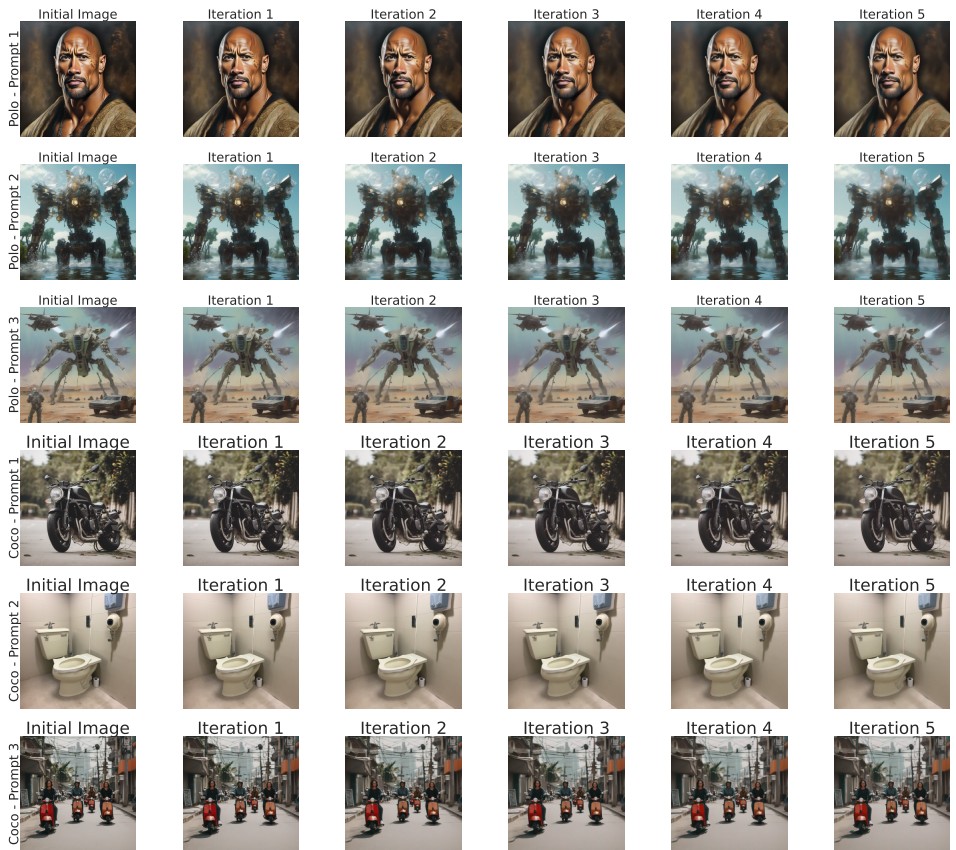

Figure 22: Initial image generation and subsequent regenerations by the SDXL1 model on Coco and Polo datasets.

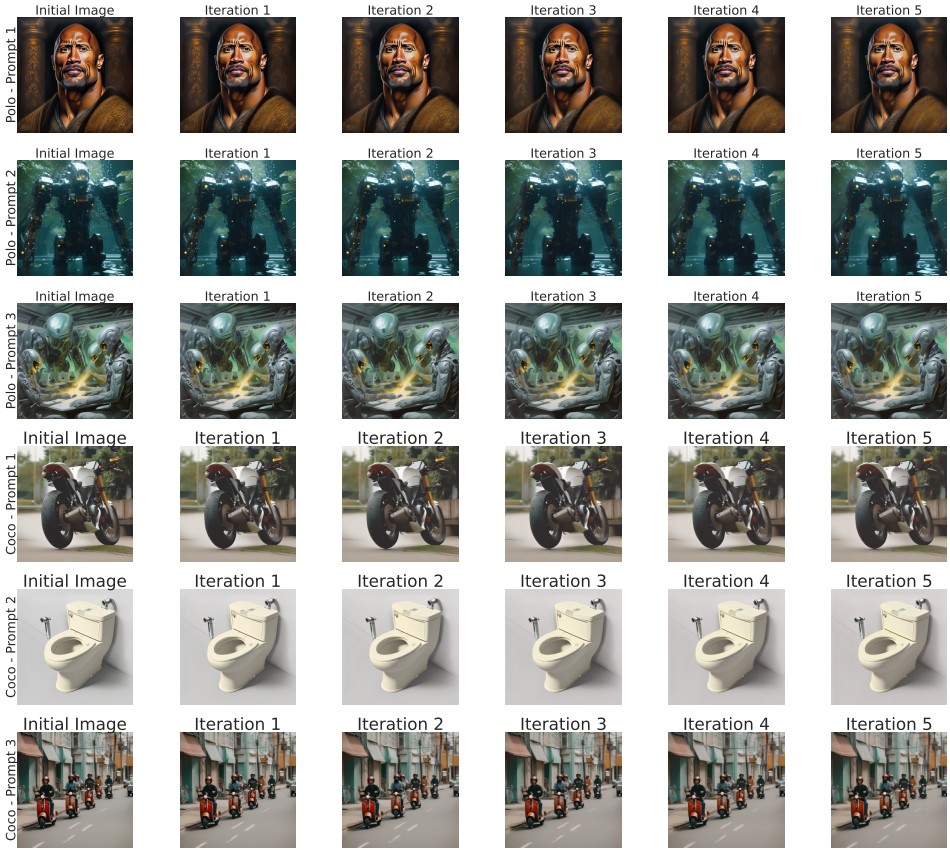

Figure 23: Initial image generation and subsequent regenerations by the SDXL0.9 model on Coco and Polo datasets.

