# OpenReview forum: "Generative Models are Self-Watermarked: Intellectual Property Declaration through Re-Generation"
_ICLR.cc/2024/Conference — Submitted to ICLR 2024_

### Official Review · Reviewer_Eg4r · 2023-10-25

**Soundness:** 2 fair
**Presentation:** 2 fair
**Contribution:** 2 fair
**Rating:** 3
**Confidence:** 4

**Summary:**

This paper utilizes the intrinsic characteristics of the generative models as the inherent fingerprints for intellectual property protection. This new method requires no addtional modifications on the training data or the model parameters. The evaluation results on both text generation and image generation tasks demonstrate the effectiveness of this method.

**Strengths:**

- Trendy topic
- New perspective for designing fingerprints
- Good verification performance

**Weaknesses:**

- Limited application scenarios
- Lack of robustness evaluation
- Exceed the page limit

**Questions:**

This paper proposes a novel fingerprinting methods via utlilizing the intrinsic characteristics of the generative models. The fingerprint difference could be strengthened by the re-generation process. The authors provide both theoretical analysis and empirical evaluations to prove the effectiveness of this method. The evaluation results on both text generation and image generation tasks demonstrate the good performance of this method.

However, I still have the following concerns.

- Though the proposed method is a good trial to implement fingerprints without any modifications on the training data or the model parameters, the application scenarios for this method are limited. This paper implicitly assumes that there is no information loss during the generation process, which allows for the multi-round re-generation process. In the experiments for text generation, the authors also merely consider the translation task which is consistent with this assumption. However, for other text generation tasks like summarization, this method might not be applicable. Moreover, even for the image generation task, the authors also implicitly assume that the model owner knows the text prompts used by the suspicious generated images. Then what if the text prompt is unknown? I would suggest the authors elaborate more on this in their paper.

- This paper lacks robustness evaluation. For instance, for the image generation task, will image transformations or pixel perturbations on the generated images affect the verification performance? I would suggest the authors add some experiments to evaluate the robustness of their method under various perturbations.

- In Table 2, when increasing $k$ from 3 to 5, the performance for the Cohere authentic model with the M2M contrast model decreases. This result is weird. I hope the authors can add some explanations here.

- The main text of this paper (before citation) has exceeded 9 pages, which violates the page limit for ICLR submission. I am afraid that this will lead to a desk rejection.

---

> ### Author Response · Authors · 2023-11-19
> **Response to Reviewer Eg4r (1/2)**
>
> **[Q1.1]**:Limited application scenarios for NLP (clarifying ‘re-generation process’, ‘translation task’ and other tasks like ‘summarization, this method might not be applicable’)
>
> **[A1.1]**: **Re-generation** can be done through various methods, including but not limited to round-trip translation and prompt-based paraphrasing. Round-trip translation is utilized as one method of re-generation step in our work, as described in Section 4.2. Furthermore, we would like to emphasize that the **target generation task** within our framework may encompass any NLG task within the scope of typical LLMs, with translation being one pertinent example.
>
> To demonstrate the generalization of our approach, we further examine two popular text generation tasks (as *target generation tasks*): paraphrasing and summarization. Upon these tasks, prompt-based paraphrasing is employed as a means of *re-generation step*. We set k to 5 because of its superior performance. The results are presented in the following tables.
>
> Paraphrasing as generation task:
> |      Models         | GPT-3.5-turbo  | | Zephyr  |  | Mistral | |
> | :---------------- | :------: | :------: |:------: | :------: |:------: | :------: |
> |  | Acc  | Mis | Acc  | Mis | Acc  | Mis |
> |GPT-3.5-turbo | - | - | 75.0 | 20.0 | 69.0 | 24.0 |
> |Zephyr | 84.0 | 10.0 | - | - | 69.0 | 23.0 |
> | Mistral | 74.0 | 19.0 | 76.0 | 10.0 | - | -|
>
> Summarization as generation task:
> |      Models         | GPT-3.5-turbo  | | Zephyr  |  | Mistral | |
> | :---------------- | :------: | :------: |:------: | :------: |:------: | :------: |
> |  | Acc  | Mis | Acc  | Mis | Acc  | Mis |
> |GPT-3.5-turbo | - | - | 92.0 | 6.0 |68.0 | 24.0 |
> |Zephyr | 88.0 | 10.0 | - | - | 67.0 | 26.0 |
> | Mistral | 86.0 | 12.0 | 86.0 | 12.0 | - | -|
>
> According to these results, our approach can effectively distinguish between authentic models and contrast models in the context of both paraphrasing and summarization (as the target tasks) with prompt-based paraphrasing (as re-generation steps).
>
> ---
>
> **[Q1.2]**: Moreover, even for the image generation task, the authors also implicitly assume that the model owner knows the text prompts used by the suspicious generated images. Then what if the text prompt is unknown?
>
> **[A1.2]**: This is a great point towards understanding our approach. The re-generation and verification do not need a task-specific prompt to regenerate the original image or to paraphrase the original text, as inpainting and paraphrasing are deterministic tasks (as outlined in Sec 4.3). Only the initial generation of images/texts may require prompts to specify the target tasks but our regeneration method does not require any prompts.
>
> ---
>
> **[Q2]**: Lacks Robustness
>
> **[A2]**: We fully recognise the robustness evaluation. As suggested, we have applied Gaussian noise on a random subset of pixels (r) on the published image before regeneration. Due to time limitations, our experiments currently utilized the Stable Diffusion v2.1 as the authentic model, however, we plan to extend our analysis to other models in our revision.  The Gaussian noise is fixed with mean (μ = 0) and large standard deviation (σ=25).
>
> Re-generation steps(k=5), Perturbation rate (r)
>
> | r                  | SDv2.1B Acc | SDv2.1B Mis | SDXL 1.0 Acc | SDXL 1.0 Mis |
> |--------------------|-------------|-------------|--------------|--------------|
> | 0%                  |  97.5        | 1.0         | 94.0         | 5.0          |
> | Gaussian Noise (0, 25) |      |             |              |              |
> | 1%               |  95.5        | 3.5         | 86.0         | 10.0         |
> | 2%               | 72.0        | 22.0        | 36.5         | 53.0         |
> | 3%               |  70.5        | 22.5        | 21.5         | 61.5         |
> | 5%               |  31.5        | 63.0          | 56.0         | 33.5         |
>
> The misclassifications in the case of a few models remain low but the accuracy notably decreases with Gaussian noise, highlighting an area for improvement. Further testing is still needed to fully evaluate verification under more adversarial conditions and other perturbation types.

---

> > ### Author Response · Authors · 2023-11-19
> > **Response to Reviewer Eg4r (2/2)**
> >
> > **[Q3]**: In Table 2, when increasing $k$ from 3 to 5, the performance for the Cohere authentic model with the M2M contrast model decreases. This result is weird. I hope the authors can add some explanations here.
> >
> > **[A3]**: We would like to highlight that the overall performance of our approach across all models improves given higher $k$ regarding both Avg Acc. and Avg Mis.
> >
> > k=3
> >
> > | Models   | Avg Acc. | Avg Mis |
> > |----------|----------|---------|
> > |M2M | 96.3 | 1.3
> > |mBART | 91.7	| 3.7 |
> > |GPT3.5-turbo	| 93.0	| 5.0 |
> > |Cohere | 70.3	| 24.0 |
> > |**Avg Acc Overall (k=3)** | 87.8 | 8.5|
> >
> > k=5
> >
> > | Models   | Avg Acc. | Avg Mis |
> > |----------|----------|---------|
> > | M2M | 95.7 | 1.3 |
> > | mBART | 93.0	| 3.3 |
> > | GPT3.5-turbo | 93.0 | 3.7|
> > | Cohere | 74.3 | 19.7|
> > |**Avg Acc Overall (k=5)** | 89.0 | 7.0|
> >
> > We attribute the cause for the performance drop in some cases to a combination of smaller distances in output and few outliers.
> >
> > ---
> >
> > **[Q4]**: Exceeding page limit
> >
> > **[A4]**: Our submission adheres to the ICLR guidelines. The exceeding part only includes Reproducibility Statement and ***“This optional reproducibility statement will not count toward the page limit”*** according to [ICLR Author Guide - Reproducibility Section](https://iclr.cc/Conferences/2024/AuthorGuide#:~:text=Reproducibility,been%20made%20to%20ensure%20reproducibility).

---

### Official Review · Reviewer_66tW · 2023-10-30

**Soundness:** 2 fair
**Presentation:** 1 poor
**Contribution:** 2 fair
**Rating:** 3
**Confidence:** 4

**Summary:**

This paper proposes a method to verify whether an image or text is generated by a protected model. This method can be useful for IP protection for the model owner.

**Strengths:**

The proposed method seems interesting. Instead of using watermarks and training-based methods, this method propose a "re-generation" strategy, which is novel to existing methods.

**Weaknesses:**

1. The paper is poorly written, which misses many necessary introductions and clarifications. For example, in the introduction section, the paper is written to be motivated to provide IP protection for the model owner. However, in the illustrative figure in Figure 1, it is more like to protect the artists (data owner)'s IP.

2. Besides, it is not clear the reason of Algorithm 2, because it is also different from the introduction in Eq. (1) and Figure 1. In Eq. (1) and Figure 1, they both use the protected model for re-generalization. Thus, there is no clues on the reason to also use $G_\times$ for re-generation.

3. For the theory part, no motivation or clarification on the meaning and use of these theories.

4. The basic assumption of the proposed method also seems problematic. For example, in the image generative models, the paper never discusses how to input the image samples to the model for re-generate. Based on my understanding, the Stable Diffusion models are text to image models. Therefore, how to input images to these models?

5. The paper should also discuss how similar or how different of the re-generation process and the traditional generation process. In practice, there is no way for the ''verifier'' to know how are the original samples are generated. In this way, whether the shift of generation process from re-generation can cause and compromise verification performance should also be discussed.

**Questions:**

Plz see the above question.

---

> ### Author Response · Authors · 2023-11-19
> **Response to Reviewer 66tW**
>
> **[Q1]**: The paper is poorly written, which misses many necessary introductions and clarifications. For example, in the introduction section, the paper is written to be motivated to provide IP protection for the model owner. However, in the illustrative figure in Figure 1, it is more like to protect the artists (data owner)'s IP.
>
> **[A1]**: There is no difference in protecting the IP by *human* or *artificial* artists in our framework. We use artefacts by Van Gogh and Picasso because i) it enhances the paper's visual appeal and ii) it circumvents potential IP infringement issues associated with commercial generative models (which is actually what we propose to protect). In addition, it is pertinent to conceptualize the artist as a *model* capable of embedding their unique style, functioning as a watermark, into their creations.
>
> ---
>
> **[Q2]**:  There are no clues on the reason to also use $G_{\times}$ for re-generation.
>
> **[A2]**:  $G_{\times}$ is used in the verification process, as it is essential to compare the distance from the authentic model $d(G_{a}(x_a), x_a)$ against those by a different (unauthentic) model $d(G_{\times}(x_a), x_a)$. This gives a margin $\delta$ to decide how confident we can claim the IP (see more discussion in Table 2).
>
> ----
>
> **[Q3]** For the theory part, no motivation or clarification on the meaning and use of these theories.
>
> **[A3]**: The motivation lies in two parts: (1) Intuitively, the authentic model should be easier to re-generate the samples previously generated by itself (formulated in Equation 1); (2) Iterative re-generation can reduce the "*self-edit*" distance (reflected by Fixed-Point Theorem in Section 3.2) and it can be utilised in verification (in Algorithm 2).
> We have highlighted and added the motivation and connection of the theories in Section 3 (see red texts). We hope our verification and answer have solved your concern.
>
> ---
>
> **[Q4]**: The paper never discusses how to input the image samples to the model for re-generate. Based on my understanding, the Stable Diffusion models are text to image models. Therefore, how to input images to these models?
>
> **[A4]**: Image re-generation is achieved by inpainting masked pixels.  The Stable Diffusion Inpainting pipeline provided by HuggingFace allows taking and masking an input image for generating specific parts of the image, conditioned on the unmasked portion of the input image. Text prompt is set to an empty string [“”] on re-generation. Please find more information about Stable Diffusion and Inpainting in HuggingFace (with an example https://huggingface.co/runwayml/stable-diffusion-inpainting). We have further revised Appendix B.1. which we hope provides more clarity. Please let us know if you would like any further clarification.
>
> ---
>
>
> **[Q5.1]**: The paper should also discuss how similar or how different the re-generation process and the traditional generation process.
>
> **[A5.1]**: We take CV tasks as an example, the target generation tasks could be any existing image generation tasks, e.g., text to image, text+image to image, etc. The re-generation process is basically generating an image `similar’ to the original input (similar to paraphrasing in NLG).
>
> **[Q5.2]**: In practice, there is no way for the ''verifier'' to know how are the original samples are generated. In this way, whether the shift of the generation process from re-generation can cause and compromise verification performance should also be discussed.
>
> **[A5.2]**:  We agree that the "verifier" may not have knowledge of the images/sentences generation process, but our verification process works on various re-generation steps $k$. Nonetheless, this study focuses on determining whether re-generation aids in the identification of IP as outlined in Section 3.1. According to Tables 2, 5, and 6, our approach demonstrates efficacy in IP identification, with further improvements achievable through iterative re-generation.
>
> ---
>
> We hope our responses have solved your concerns. We are pleased to answer additional questions.

---

> ### Comment · Reviewer_66tW · 2023-11-21
>
> I acknowledge that I receive and thank for the authors' response.
>
> Although the authors provide brief clarifications, I could still not clearly understand the basic problem setup and detailed algorithm. In detail, it is not clear that the protection subject is the model or the user's data, although the authors claim there is not difference between these two. I suggest the authors provide explicit problem definition before introducing the solutions in the future versions.
>
> Besides, the theorems and algorithms are still briefly introduced. For example, the "re-generation" process is a very important component in the algorithm, and it could still be very different in different generative models. However, a detailed introduction is not provided.
>
> Based on these reasons, I tend to keep my original rating.

---

> ### Author Response · Authors · 2023-11-22
> **Clarification**
>
> Thank you for your reply. We want to clarify that:
> 1. Our framework can authenticate the IP of data samples produced by either AI models or human contributors. The crux of this verification lies in ascertaining that the IP of the data is a direct result of the IP inherent in the models or the human creators involved. To illustrate, consider the example of authoring a book. The fundamental aim in such a scenario is to assert the IP of the book, underscoring that it is human’s or model’s creation. This analogy serves to explain our primary goal in this research.
> 2. The re-generation process is as simple as $x'=f(x; \theta)$, where $f(\cdot; \theta)$ is a `paraphrase-style` or `inpainting-style` generator. It is widely supported by AGI models such as ChatGPT, Stable Diffusion and their successors. The detailed descriptions are provided in Sections 4.2 and 4.3.
> 3. Our method gives insight in how to enhance and verify the IP by using the simplest re-generation process without (1) manipulating the generative models and their outputs or (2) additional classification models which require training.

---

### Official Review · Reviewer_Bepi · 2023-11-01

**Soundness:** 3 good
**Presentation:** 3 good
**Contribution:** 2 fair
**Rating:** 5
**Confidence:** 3

**Summary:**

The paper introduces a method for verifying data authorship in generative models without conventional watermarking. It discovers latent fingerprints inherent in deep learning models by comparing original data samples with their re-generated counterparts. It introduces a framework for generating and verifying these model fingerprints and establishes a practical protocol to authenticate data ownership in generative models.

**Strengths:**

1. The approach of uncovering latent fingerprints through regeneration is novel. The observation of distance convergence is straightforward and interesting.

2. The method is easy to implement. It uses standard generative models without requiring complex modifications to the model or additional tools.

**Weaknesses:**

1. In the neural language generation experiments, the paper used round-trip translation as re-generation. Since this is a relatively easy task, the same method might not generalize well to other, more demanding text generation tasks, such as creative writing which is more prevalent in real-world applications.

2. The method’s design, focused on distinguishing outputs between specific models, may not be adept at confirming if a piece of content is exclusively generated by a particular model. This is especially pertinent in scenarios where it’s critical to ascertain whether content originated from a specific source, and not just differentiate it from another known model.

3. The method is less effective for certain models as shown in Figure 3 and Appendix D.2.

4. The theoretical analysis looks good. One concern is that the Lipschitz constant could be larger than 1 for complex/deep models. Thus, the assumption may not hold for real-world generative models. This is not a major issue. Could authors mention this?

5. The description of the method is not very clear. The subsection 3.3 is very short, which makes it very hard to understand. I suggest the authors to give some examples when describing the method. Otherwise, it is very challenging to understand the method after reading Section 3.

6. The cost is very large because we need to repeat it k times to generate an output. Also, from Section 3.3, it is not clear how exactly the re-generation is done for text and image, e.g., line 33 in Algorithm 1.

7. The robustness is not considered, e.g., an attacker could slightly change the words in a text. Will the proposed method still be effective in this case? I would suggest authors to conduct some adaptive attacks to study the robustness of the verification.

**Questions:**

See above.

---

> ### Author Response · Authors · 2023-11-19
> **Response to Reviewer Bepi (1/2)**
>
> Thank you for appreciating the novelty and simplicity of our framework.
>
> ---
>
> **[Q1]**:  Round-trip translation is relatively easy,  the same method might not generalize well to others.
>
> **[A1]**: **Re-generation** can be done through various methods, including but not limited to round-trip translation and prompt-based paraphrasing. Round-trip translation is utilized as one method of re-generation step in our paper, as described in Section 4.2. Furthermore, we would like to emphasize that the **target generation task** within our framework may encompass any task within the scope of LLMs, with translation being one pertinent example.
>
> To demonstrate the generalization of our approach, we further examine two popular text generation tasks using prompting: *paraphrasing* and *summarization*. In these tasks, *prompt-based paraphrasing* is employed as the re-generation step. We set k to 5 because of its superior performance. The results are presented in the following tables.
>
>
> Paraphrasing as generation task:
>
>  |      Models         |  **GPT-3.5-turbo**  | | **Zephyr**  |  | **Mistral** | |
> | :---------------- | :------: | :------: |:------: | :------: |:------: | :------: |
> |  | Acc  | Mis | Acc  | Mis | Acc  | Mis |
> |GPT-3.5-turbo | - | - | 75.0 | 20.0 | 69.0 |24.0 |
> |Zephyr | 84.0 | 10.0 | - | - | 69.0 | 23.0 |
> | Mistral | 74.0 | 19.0 | 76.0 | 10.0 | - | -|
>
>
> Summarization as generation task:
> |      Models         | **GPT-3.5-turbo**  | | **Zephyr**  |  | **Mistral** | |
> | :---------------- | :------: | :------: |:------: | :------: |:------: | :------: |
> |  | Acc  | Mis | Acc  | Mis | Acc  | Mis |
> |GPT-3.5-turbo | - | - | 92.0 | 6.0 |68.0 |24.0 |
> |Zephyr | 88.0 | 10.0 | - | - | 67.0 | 26.0 |
> | Mistral | 86.0 | 12.0 | 86.0 | 12.0 | - | -|
>
> According to these results, our approach can effectively distinguish between authentic models and contrast models in the context of both paraphrasing and summarization (as the target tasks) with prompt-based paraphrasing (as re-generation steps).
>
> ---
>
> **[Q2]**: The method may not reliably confirm the source model, since it focuses on differentiating between models rather than identifying unique sources.
>
> **[A2]**: We would like to emphasize that the proposition of IP detection is typically initiated by the model owners. This means that should the owners suspect an IP infringement, they can utilize our approach to substantiate that the images or sentences in question are outputs of their authentic models, rather than those generated by any other models (claimed by the opposing party). This assertion has been corroborated through our experimental findings, as illustrated in Figures 3 and 4. Consequently, we contend that our approach is efficacious in asserting IP claims.
>
>
> ---
>
> **[Q3]**: The method is less effective for certain models as shown in Figure 3 and Appendix D.2.
>
> **[A3]**: We acknowledge the question regarding the effectiveness of our method on NLP models. Firstly, M2M, mBART and GPT are clearly distinguished from each other and such a conclusion is also supported in Tables 2 and 7. Secondly, different versions of GPT (1) are slightly more challenging to differentiate and (2) have no requirement of discrimination as they belong to the same company.

---

> ### Author Response · Authors · 2023-11-19
> **Response to Reviewer Bepi (2/2)**
>
> **[Q4]**: One concern is that the Lipschitz constant could be larger than 1 for complex/deep models. Thus, the assumption may not hold for real-world generative models. This is not a major issue. Could authors mention this?
>
> **[A4]**: We add an empirical estimation of the Lipschitz constant $L$ of Stable Diffusion models. We use all pairs of images ($x$ and $y$) from Polo dataset, representing them as inputs and applying the regeneration function $f(\cdot)$. We measure their distances using $\mathbb D$, i.e. Euclidean and LPIPS. The ratio below is the transformation of Equation 4:
> $$L \triangleq \frac{D(f(x), f(y))}{D(x,y)}$$
>
> | Model     | **LPIPS** | **(L)** | **Euclidean** | **(L)** |
> |-----------|--------------|-------------|------------------|-----------------|
> || Mean | Std | Mean | Std |
> | SD v2.1   | 0.976        | 0.015       | 1.000            | 0.001           |
> | SDXL 1.0  | 0.943        | 0.21        | 1.000            | 0.004           |
> | SD v2     | 0.975        | 0.016       | 1.000            | 0.001           |
> | SD v2.1B  | 0.970        | 0.023       | 1.000            | 0.002           |
> | SDXL 0.9  | 0.946        | 0.020       | 0.999            | 0.005           |
>
> Our analysis adds several insights to our work:
> 1. The mean L values are consistently less than for LPIPS, which explains the superior performance using LPIPS and the reason that Euclidean distance does not work in our study.
> 2. Usually, better (latest) models exhibit better (lower) L constant.
>
> Note that our framework focuses on empowering the watermarking property for models that **CAN** be verified. For those models that currently have not worked under our framework, we suggest targeting better model performance considering our observation and analysis (item 2).
>
> We believe these additional experiments would add confidence to the assumption used in the theory.
>
> ---
>
> **[Q5]**: The subsection 3.3 is very short, which makes it very hard to understand. I suggest the authors give some examples when describing the method. Otherwise, it is very challenging to understand the method after reading Section 3.
>
> **[A5]**: Thank you for your suggestion on the clarity of our methodology section. Imagine artists refining their distinct writing or painting styles during each artwork replication. Similarly, a generative model's unique `style' becomes more defined during image re-generation, as deviations reduce. This mirrors iterative functions which converge to fixed points. In our case, each re-generation brings the image closer to the model's inherent style, and the distinct fingerprint facilitates AI authorship verification. We have incorporated this clarification into our revision (please see the red text pieces in Section 3.3).
>
> ---
>
> **[Q6]**: The cost is very large because we need to repeat it k times to generate an output.
>
> **[A6]**: We believe the cost is affordable for the additional watermark property.
>
> 1. The cost is arguably not *very* large, as it is linearly correlated to the original target generative task, i.e. $\Theta(T)$ to $\Theta(kT)=\Theta(T)$ where k is a constant and our watermark is empirically effective when $k \in [1, 5]$.
> 2. It is worth noting that the samples can be verified by even 1-step re-generation, see Tables 2, 6 and 7 (k=1).
>
> ---
> **[Q7]**: The robustness is not considered, e.g., an attacker could slightly change the words in a text. Will the proposed method still be effective in this case?
>
> **[A7]**: Thank you for the suggestion! We have added experiments to test the robustness against ‘slightly change the words’.  We conduct perturbation (i.e. replacing some words with random ones) on each input prior to a one-step re-generation process. The perturbation rate is varied ranging from 10% to 50%. Due to the time limitation, our experiments currently utilise the GPT-3.5-turbo as the authentic model (therefore the corresponding column is left empty), however, we plan to extend our analysis to other models in our revision.
>
> |      k=5         | M2M  | | mBART  |  | GPT3.5-|turbo | Cohere | |
> | :---------------- | :------: | :------: |:------: | :------: |:------: | :------: |:------: | :------: |
> | Perturbation rate (r) | Acc  | Mis | Acc  | Mis | Acc  | Mis |Acc  | Mis |
> |0% | 90.0  | 6.0 | 94.0 | 2.0 | -  | - | 95.0  |3.0|
> |10% | 91.0 | 6.0 | 91.0 | 3.0 | -  | - | 97.0 | 2.0|
> | 20% | 84.0 | 12.0 | 84.0 | 11.0 | -  | - | 95.0 | 4.0|
> |30% | 84.0 | 12.0 | 74.0 | 15.0 | - | - | 94.0 | 3.0|
> | 40% | 79.0 | 17.0 | 75.0 | 16.0 | - | - | 93.0 | 5.0|
> | 50% | 69.0 | 22.0 | 65.0 | 22.0 | -  | - | 92.0 | 6.0 |
>
> According to the Table above, as the perturbation rate increases, there is a corresponding gradual decrease in the accuracy of our method. The overall capability of watermark verification is reasonably positive. Nonetheless, we argue that 30-50% perturbation has already significantly harmed the quality of the generated texts and attackers can hardly get much benefit from this.

---

### Official Review · Reviewer_XHuY · 2023-11-01

**Soundness:** 2 fair
**Presentation:** 3 good
**Contribution:** 2 fair
**Rating:** 5
**Confidence:** 3

**Summary:**

The topic of intellectual property in Generative Adversarial Networks has been studied for quite some time. In this paper, the authors introduce an iterative re-generation method to enhance the fingerprints within current state-of-the-art generative models. The paper is well-organized and easy to follow, with a simple yet effective main contribution. The proposed method can be easily implemented to protect intellectual property without requiring any white-box settings, making it a lightweight and easily verifiable solution.

**Strengths:**

+ The proposed method is easy yet effective. The experiments are conducted comprehensively on both NLP and CV generative models. The proposed method can be easily implemented to protect intellectual property without requiring any white-box settings, making it a lightweight and easily verifiable solution.

**Weaknesses:**

- I am still unclear about how to verify the convergence of the distance between one-step re-generation (Distance Convergence). In Equation 2, the distance converges to 0, given that L ∈ (0, 1). However, how can we guarantee that L is less than 1? Is this only confirmed through experiments?

**Questions:**

Is the value of \epsilon sensitive in the experiments? Since many LLM or MLM models are black-box, how can one select a sensible \epsilon?
In the Appendix, as k increases, the accuracy does not seem to increase significantly. The experimental results may not be entirely consistent with the analyses in Equation 2. Is this discrepancy due to the presence of some other bad case samples, or is it because the distances are not larger enough?

---

> ### Author Response · Authors · 2023-11-19
> **Response to Reviewer XHuY (1/2)**
>
> We would like to take this opportunity to highlight some significance of our work compared with previous work.
> 1. **[Model performance]**: Our approach does not rely on additional post-processing and model training/manipulation, which means no harm to the performance of the generative models.
> 2. **[Verification Advantage]**: Our approach is independent of additional classifiers for verification. This means it avoids the potential robustness issues of domain adaptation for classifiers, considering verifying samples generated by new models (new company, new training dataset, new architecture etc.).
> 3. **[Scenario]**: (a) In future, generative models are not required to add additional watermarks for verification. (b) For previously published AI-generated samples, our approach manages to trace back the IP infringement (considering many generative models in various applications have not yet seriously considered embedding watermarks to their systems).
> 4. **[Simplicity]**: Our approach is simple and effective. It means many generative AI models, CV and NLP as examples in our paper, can benefit from our research.
>
> ---
>
> **[Q1]**: how to verify the convergence of the distance between one-step re-generation (Distance Convergence)
>
> **[A1]**: As demonstrated in Figures 2 and 4, the one-step re-generation distance will converge regarding various metrics for both NLP and CV generative models. In Algorithm 1, the generator has already finished $k$ steps re-generation, $k \in \mathbb Z^{+}$, and the distance derived in the verification process (in Algorithm 2) is equivalent to the $k+1$-th step. Therefore, the $k+1$-th step re-generation distances (in verification) theoretically converge to small values and be empirically verifiable.
>
>
> ---
>
>
> **[Q2]**: How can we guarantee that L is less than 1?
>
> **[A2]**: Thank you for your question. We add an empirical estimation of the Lipschitz constant $L$ of Stable Diffusion models. We use all pairs of images ($x$ and $y$) from Polo dataset, representing them as inputs and applying the regeneration function $f(\cdot)$. We measure their distances using $\mathbb D$, i.e. Euclidean and LPIPS. The ratio below is the transformation of Equation 4:
> $$L \triangleq \frac{\mathbb D(f(x), f(y))}{\mathbb D(x,y)}.$$
>
>
> | Model     | L_LPIPS Mean | L_LPIPS Std | L_Euclidean Mean | L_Euclidean Std |
> |-----------|--------------|-------------|------------------|-----------------|
> | SD v2.1   | 0.976        | 0.015       | 1.000            | 0.001           |
> | SDXL 1.0  | 0.943        | 0.21        | 1.000            | 0.004           |
> | SD v2     | 0.975        | 0.016       | 1.000            | 0.001           |
> | SD v2.1B  | 0.970        | 0.023       | 1.000            | 0.002           |
> | SDXL 0.9  | 0.946        | 0.020       | 0.999            | 0.005           |
>
> Our analysis adds several insights to our work:
> 1. The mean L values are consistently less than for LPIPS, which explains the superior performance using LPIPS and the reason that Euclidean distance does not work in our study.
> 2. Usually, better (latest) models exhibit better (lower) L.
>
> It is worth noting that our framework focuses on empowering the watermarking property for models that **CAN** be verified, instead of enforcing that every model is verifiable. For those models that currently have not worked under our framework, we suggest targeting better model performance considering our observation and analysis (item 2).
>
> ---
>
>
> **[Q3]**: Sensitivity of $\epsilon$
>
> **[A3]**: $\epsilon$ has not been employed  in our paper. We assume you were inquiring about the $\delta$, a critical hyper-parameter used in Algorithm 2. We set $\delta$ to 0.05, as it yields optimal results for NLP experiments according to Table 1. We observe the same trend for CV experiments below.
>
> The accuracy of differentiating contrast models from the authentic model SDv2.1 using different $\delta$.
>
> | $\delta$ | SD v2.   | SD v2.1B | SDXL 0.9 | SDXL 1.0 |
> |----------|----------|----------|----------|----------|
> | 0.05     | 0.990    | 1.00     | 1.00     | 1.00     |
> | 0.10     | 0.985    | 1.00     | 1.00     | 1.00     |
> | 0.20     | 0.960    | 1.00     | 1.00     | 1.00     |
> | 0.40     | 0.895    | 1.00     | 0.995    | 0.985    |

---

> ### Author Response · Authors · 2023-11-19
> **Response to Reviewer XHuY (2/2)**
>
> **[Q4]**: In the Appendix, as k increases, the accuracy does not seem to increase significantly. The experimental results may not be entirely consistent with the analyses in Equation 2. Is this discrepancy due to the presence of some other bad case samples, or is it because the distances are not large enough?
>
> **[A4]**: We assume you are referring to Table 5 regarding the minimal performance increase when k increases. We would like to highlight that the overall performance of our approach across all models improves given higher $k$ regarding both Avg Acc. and Avg Mis. The row average and overall average scores for Table 2 and Table 5 are calculated and presented as follows:
>
> k=3:
>
> | Models   | Avg Acc. | Avg Mis. |
> |----------|----------|---------|
> | SDv2.1   | 90.125   | 7.875   |
> | SDv2     | 89.625   | 7.625   |
> | SDv2.1B  | 78.000   | 17.50   |
> | SDXL 0.9 | 90.750   | 8.000   |
> | SDXL 1.0 | 93.250   | 4.875   |
> | **Overall Avg** | **88.350** | **9.175** |
>
> k=5:
>
> | Models   | Avg Acc. | Avg Mis. |
> |----------|----------|---------|
> | SDv2.1   | 88.5     | 8.625   |
> | SDv2     | 88.375   | 8.625   |
> | SDv2.1B  | 80.875   | 14.375  |
> | SDXL 0.9 | 96.625   | 2.25    |
> | SDXL 1.0 | 93.125   | 5.25    |
> | **Overall Avg** | **89.5** | **7.825** |
>
>
> NLP:
> k=3
> | Models   | Avg Acc. | Avg Mis. |
> |----------|----------|---------|
> |M2M | 96.3 | 1.3
> |mBART | 91.7	| 3.7 |
> |GPT3.5-turbo	| 93.0	| 5.0 |
> |Cohere | 70.3	| 24.0 |
> |**Overall Avg** | **87.8** | **8.5** |
>
> k=5
>
> | Models   | Avg Acc. | Avg Mis. |
> |----------|----------|---------|
> | M2M | 95.7 | 1.3 |
> | mBART | 93.0	| 3.3 |
> | GPT3.5-turbo | 93.0 | 3.7|
> | Cohere | 74.3 | 19.7|
> |**Overall Avg** | **89.0** | **7.0** |
>
> ---
> We hope our clarification and additional experiments have solved your concerns. We are pleased to take additional questions.

---

### Meta-Review · Area_Chair_sogc · 2023-12-05

**Metareview:**

This paper is working on protecting intellectual property for generated data. Authors proposed a verification procedure to verify data ownership through re-generation. They also introduce a method to amplify the model fingerprints through iterative data re-generation and a theoretical grounding on the proposed approach. Authors demonstrated their work for both text and image generative models. Strengths of this paper are: 1) proposed method is easy, could protect IP without white-box setting; 2) the approach of uncovering latent fingerprints through regeneration is novel. Weaknesses are: 1) the method is less effective for certain models; 2) authors conducted experiments for a pairs of models, lack of results for multiple models; 3) lacks comparison with simple classification based methods; 4) more ablation studies or experimental results needed to verify the effectiveness of the proposed method. This paper got 2 "5: marginally below the acceptance threshold" and 2 "3: reject, not good enough" rating. During discussion only reviewer 66tW replied and didn't convinced by authors' reply. Other reviewers kept their rating.

**Justification For Why Not Higher Score:**

Reviewers' concerns are not well addressed. This paper still need some polish and more experimental results are needed.

**Justification For Why Not Lower Score:**

N/A

---

### Decision · Program_Chairs · 2024-01-16

Reject